# Rectifying Soft-Label Entangled Bias in Long-Tailed Dataset Distillation

**Chenyang Jiang[1,2]**    **Hang Zhao[1]**    **Xinyu Zhang[1]**    **Zhengcen Li[1,2]**
**Qiben Shan[2]**    **Shaocong Wu[2]***    **Jingyong Su[1,2]***

[1]Harbin Institute of Technology, Shenzhen    [2]Pengcheng Laboratory

{23B936035,200110431,zhangxinyu,lizhengcen}@stu.hit.edu.cn
sujingyong@hit.edu.cn, {shanqb, wushc}@pcl.ac.cn

## Abstract

Dataset distillation compresses large-scale datasets into compact, highly informative synthetic data, significantly reducing storage and training costs. However, existing research primarily focuses on balanced datasets and struggles to perform under real-world long-tailed distributions. In this work, we emphasize the critical role of soft labels in long-tailed dataset distillation and uncover the underlying mechanisms contributing to performance degradation. Specifically, we derive an imbalance-aware generalization bound for model trained on distilled dataset. We then identify two primary sources of soft-label bias, which originate from the distillation model and the distilled images, through systematic perturbation of the data imbalance levels. To address this, we propose ADSA, an Adaptive Soft-label Alignment module that calibrates the entangled biases. This lightweight module integrates seamlessly into existing distillation pipelines and consistently improves performance. On ImageNet-1k-LT with EDC and IPC=50, ADSA improves tail-class accuracy by up to 11.8% and raises overall accuracy to 41.4%. Extensive experiments demonstrate that ADSA provides a robust and generalizable solution under limited label budgets and across a range of distillation techniques.

## 1 Introduction

Pretrained models built on large-scale datasets exhibit superior predictive capabilities. However, training such large-scale models demands massive data volumes, resulting in significant costs in data storage, transmission, and computational resources. Moreover, the effectiveness of model training is highly dependent on the availability of high-quality datasets [1, 2]. With the increasing importance of data, the paradigm of data-centric artificial intelligence (DCAI) has gained increasing attention, encompassing data acquisition, optimization, and maintenance [3], and has become central to contemporary model development in both research and industry. A key research challenge is to efficiently extract critical information from datasets, eliminate redundancy, and construct high-quality data, which can accelerate model development, reduce carbon emissions, and promote the democratization of large-scale model training.

Dataset distillation [4] aims to synthesize a smaller and high-quality synthetic dataset that serves as a surrogate for the original dataset. The core motivation is to treat the dataset as learnable parameters by representing images as trainable tensors and optimizing them to match the performance of models trained on the full dataset. Recent advances in dataset distillation have focused on improved optimization objectives [5], parameterization strategies [6], training dynamics [7], and more effective utilization of soft labels [8], which has significantly enhanced its practicality and applicability.

---

*Corresponding authors.

Despite recent progress in dataset distillation, limited attention has been paid to the challenge of distilling long-tailed datasets, a pervasive issue in real-world scenarios. Real-world data typically follow a long-tailed distribution [9, 10], where a few head classes account for the majority of samples, while numerous tail classes are sparsely represented. Models trained on such distributions often suffer significant performance degradation on tail classes. Applying dataset distillation directly to these datasets tends to produce biased synthetic data, leading to performance degradation and optimization instability. This hinders the real-world applicability of dataset distillation. Such scenarios often involve a distributional mismatch between the original and synthetic datasets. LTDD [11] attempts to address this issue by aligning the parameter distributions derived from models trained on both datasets during the distillation process. While LTDD provides a promising solution by aligning parameter distributions, it primarily operates at the parameter-level matching and does not explicitly address the bias encoded in soft labels, an increasingly critical component in modern distillation pipelines.

Our work focuses on how soft labels, a widely used technique in recent dataset distillation methods [12, 13, 14, 15, 16] that significantly boosts performance, are affected by dataset imbalance. We first derive an imbalance-aware generalization bound for dataset distillation, which theoretically reveals that imbalanced soft labels generated during the standard distillation process limit the performance of models trained on the distilled data. We then conduct an experiment that perturbs the distribution of the original dataset and observe its effect on the distilled images and labels. Through this analysis, we identify two primary sources of soft-label bias: the distillation model and the distilled images. These biases jointly contribute to significant performance degradation on tail classes.

To address the entangled bias inherent in soft labels, we introduce ADSA, an ADaptive Soft-label Alignment module that calibrates the distillation model's predictions on distilled images by jointly removing both sources of bias. We observe that distilled images can serve as a form of hold-out data to estimate and eliminate this bias. ADSA preserves the class-level relational information encoded in soft labels while eliminating the bias induced by the imbalanced distillation model and the image-specific distribution shift. It functions as a post-hoc module that does not participate in model training or image distillation, and can therefore be seamlessly integrated into various dataset distillation baselines, without requiring the design of complex training objectives, sampling strategies, or distillation architectures to handle long-tailed distributions. Moreover, it can benefit from advanced methods for distilling imbalanced datasets at the ImageNet scale. This lightweight module is simple yet effective, and consistently improves accuracy across multiple state-of-the-art dataset distillation methods and diverse datasets, providing a robust and generalizable solution for imbalanced data scenarios across various distillation settings.

Our main contributions are summarized as follows:

- We theoretically and empirically reveal the importance of soft labels in dataset distillation by deriving an imbalance-aware generalization bound and designing experiments to investigate the influence of imbalanced data. The theoretical analysis and corresponding experiment design provide an effective tool for future research on dataset distillation under distribution shift.
- We reveal two distinct sources of soft-label bias originating from the imbalanced distillation model and the image-induced distribution shift, both of which jointly degrade tail-class performance.
- We propose ADSA, an adaptive soft-label alignment module that calibrates soft-label distributions in a post-hoc manner. ADSA is lightweight, plug-and-play, and can be seamlessly integrated into various dataset distillation methods without the need for complex training design.

## 2 Related Work

Dataset distillation reduces dataset size while preserving essential information, making it valuable for neural network training. It has been applied in real-world scenarios, such as continual learning. However, most real-world datasets exhibit a long-tail distribution, and research on dataset distillation in such contexts remains limited. This section first reviews key studies on dataset distillation, followed by an overview of long-tail recognition and relevant research.

### 2.1 Dataset Distillation

Dataset distillation [4] aims to condense the knowledge of an original dataset into a significantly smaller synthesized dataset. This process involves initializing the synthetic dataset as trainable

parameters and optimizing it using gradients derived from a specified objective function. The objective function is a carefully designed matching criterion between the core information of the original and synthesized datasets, where an information function extracts essential features, and a distance function computes the loss. This information extraction process typically involves model training or inference, where the model is referred to as a *distillation model*. Finally, we evaluate performance by training an *evaluation model* on the distilled dataset and testing it on the test set. Methods are categorized into performance matching, gradient matching, parameter matching, and distribution matching based on the matching policy.

Performance matching [4, 17, 18] optimizes the performance of model trained on the distilled dataset directly. Gradient matching [19, 20, 21, 22] and parameter matching [23, 24, 7] both train a distillation model on the distilled dataset to mimic the behavior of model trained on the original dataset, with the former focusing on gradient similarity and the latter on training parameter trajectory similarity.

The matching methods described above are challenging to deploy in large-scale dataset scenarios due to the high computational and memory costs associated with distillation model training and loading in the inner loop of the distillation process. Consequently, distribution-based methods were initially proposed by Bo Zhao [25], which optimize the synthesized dataset by matching the feature distributions of neural networks. DataDAM [26] and DREAM [27] improved the matching and sampling strategies to enhance performance. More recently, SRe2L [14] further boosted the performance of distribution-based methods by utilizing a DeepInversion-like [28] image distilling approach and soft labels, establishing a three-stage framework of squeeze, recover, and relabel. Subsequently, D3S [29], EDC [16], and GVBSM [30] enhanced performance both theoretically and empirically. In addition to optimization-based methods, generative-based approaches have also been explored. These methods conduct the matching process in the latent space [31, 15] or modify the generative models' objective function to encourage the generation of more informative images [32].

Soft labels for distilled images have been widely adopted by many methods [12, 13, 14, 15, 16] to enhance performance, and have recently received increasing attention. Qin et al. [33] highlight the crucial role of soft labels in dataset distillation and systematically evaluate their impact on model performance. The DD-Ranking benchmark [34] further demonstrates that dataset distillation techniques can lead to substantial performance improvements. GIFT [8] introduces a tailored loss function for training on distilled datasets to better exploit soft labels. Xiao et al. [35] find that high within-class diversity demands large-scale soft labels and propose the LPLD method to effectively prune them. The optimization of soft labels has also been explored in prior work [36, 37], while DRUPI [38] and LADD [39] design more effective soft-label formats to improve training efficiency.

Recent advancements have brought data distillation closer to real-world applications, where data typically follows a long-tailed distribution. Directly applying distillation techniques to long-tailed datasets may result in a biased distilled dataset. LTDD [11] pioneers long-tailed dataset distillation by identifying the limitations of expert trajectory matching on imbalanced data and proposing Weight Mismatch Avoidance and Adaptive Decoupled Matching to improve tail-class supervision and soft-label quality. However, its focus remains on parameter-level bias propagation, while our work explicitly targets soft-label bias and introduces a post-hoc calibration strategy to correct it.

## 2.2 Long-Tailed Recognition

Long-tailed recognition aims to train well-performing deep models, particularly on tail classes, from datasets following a long-tailed class distribution [9]. Traditional designs of sampling methods, network architecture, and loss functions cause the trained network to assign much higher confidence to the head classes than to the tail classes, leading to poor predictions on the tail classes [40]. Balanced data acquisition [41], multi-branch networks [42], feature transferring [43, 44, 45], adaptive loss function design [46, 47], decoupled training [48], and logit post-hoc calibration [49] have been well-explored. Decoupled training methods first train the model's backbone under standard settings and then apply balanced sampling to retrain the classifier, achieving significant performance gains. This reveals that the backbone trained on a long-tailed dataset retains sufficient capacity to extract tail-class information. Our approach takes this further by utilizing a fully trained distillation model without modification to recover tail-class information within the distilled dataset, thereby enhancing the evaluation model's training.

## 3 Method

### 3.1 Imbalance-aware Upper Bound

Dataset distillation aims to distill the original dataset $D_{tr}$ into a smaller synthetic dataset $D_{dd}$, where the size of the latter is significantly smaller than the former, i.e., $|D_{dd}| \ll |D_{tr}|$. We define the loss function for model $f_\theta$ (hereafter referred to $\theta$ for brevity) and dataset $D$ as $l(\theta, D) = \frac{1}{|D|} \sum_{(x,y) \in D} L(f_\theta(x), y)$, where $L$ denotes the cross-entropy function. The goal of dataset distillation is to minimize the performance gap on the test dataset $D_{te}$ between models $\theta_{tr}$ and $\theta_{dd}$, trained on $D_{tr}$ and $D_{dd}$ respectively:

$$D_{dd} = \underset{D_{dd}}{\operatorname{argmin}} \left[ l(\theta_{dd}, D_{te}) - l(\theta_{tr}, D_{te}) \right] \tag{1}$$

$$\theta_{dd} = \underset{\theta}{\operatorname{argmin}} \, l(\theta, D_{dd}). \tag{2}$$

For theoretical analysis, it is common to model the finite datasets $D_{tr}$ and $D_{dd}$ as samples drawn from underlying distributions, $p_{tr}(x, y)$ and $p_{dd}(x, y)$, respectively. The empirical loss $l(\theta, D)$ is thus a Monte Carlo approximation of the expected loss, $\mathbb{E}_p[L(f_\theta(x), y)]$. Following this convention, we adopt the notation for expected losses in our theoretical discussion.

The theoretical framework of D3S [29] views dataset distillation as a domain shift problem. For a classifier $\hat{p}(y|x)$ trained on data from $p_{dd}$, its corresponding expected losses are defined as $l_{dd} = \mathbb{E}_{p_{dd}}[-\log \hat{p}(y|x)]$ and $l_{tr} = \mathbb{E}_{p_{tr}}[-\log \hat{p}(y|x)]$. Assuming the classifier's negative log-likelihood is bounded by a positive constant $C$, D3S provides the following generalization bound:

$$l_{tr} \leq l_{dd} + \frac{C}{2\sqrt{2}} \sqrt{R_{dd}}, \tag{3}$$

$$\text{where} \quad R_{dd} = D_{KL}(p_{tr}(x) \| p_{dd}(x)) + D_{KL}(p_{tr}(y|x) \| p_{dd}(y|x)). \tag{4}$$

This result suggests that the upper bound on the original training loss $l_{tr}$ can be tightened by reducing the training loss $l_{dd}$ on the distilled dataset and minimizing the distribution discrepancy $R_{dd}$ between the original training and distilled datasets. Under $p_{tr}(x, y) = p_{te}(x, y)$ assumption and with a sufficiently large dataset, we have $l_{te} \approx l_{tr}$ according to VC theory [50]. Consequently, the test loss $l_{te}$ can be considered approximately upper-bounded by $l_{dd} + \frac{C}{2\sqrt{2}} \sqrt{R_{dd}}$. However, this assumption does not hold in the long-tailed setting, where only the class-conditional distributions $p_{tr}(x|y) = p_{te}(x|y)$ are preserved in long-tailed recognition studies [51, 46].

Following the setting in long-tailed recognition study [51, 46], the training dataset $D_{dd}$ used for distillation follows a long-tailed distribution, while the test dataset is balanced. Let $n_k$ denote the number of samples in the $k$-th class in $D_{tr}$, where $\sum_{k=0}^{K-1} n_k = |D_{tr}|$ and $n_0 > n_1 > \cdots > n_{K-1}$.

**Theorem 3.1.** *In the long-tailed setting, where training and test distributions share the same class-conditional distributions (i.e., $p_{tr}(x|y) = p_{te}(x|y)$) but differ in their class priors (i.e., $p_{tr}(y) \neq p_{te}(y)$), the discrepancy term $R_{dd}$ in the D3S bound* (4) *can be expressed in the following two equivalent forms:*

$$R_{dd} = D_{KL}(p_{te}(y|x) \| p_{dd}(y|x)) + D_{KL}(p_{te}(x) \| p_{dd}(x)) + const, \tag{5}$$

*and*

$$R_{dd} = D_{KL}(p_{te}(y) \| p_{dd}(y)) + \sum_y p_{te}(y) D_{KL}\left(p_{tr}(x|y) \| p_{dd}(x|y)\right). \tag{6}$$

The proof is provided in Appendix A.2. Under the relaxed assumption, the two resulting bounds introduce additional terms compared to Eq. 4, leading to several key insights regarding the desired properties of the distilled dataset: (i) The first term in Eq. 5 suggests that the learned posterior distribution $p_{dd}(y|x)$ from a model trained on $D_{dd}$ should align with the label distribution in the test set. (ii) The second term in Eq. 5 highlights the importance of aligning the feature distribution of the synthetic dataset with that of the original training data to ensure optimal performance. Additionally, the second term in Eq. 6 refines this insight by emphasizing a more fine-grained, per-class alignment due to the long-tailed assumption $p_{tr}(x|y) = p_{te}(x|y)$. (iii) The first term in Eq. 6 further suggests a natural experimental setup by adopting a balanced class distribution in the distilled dataset, where

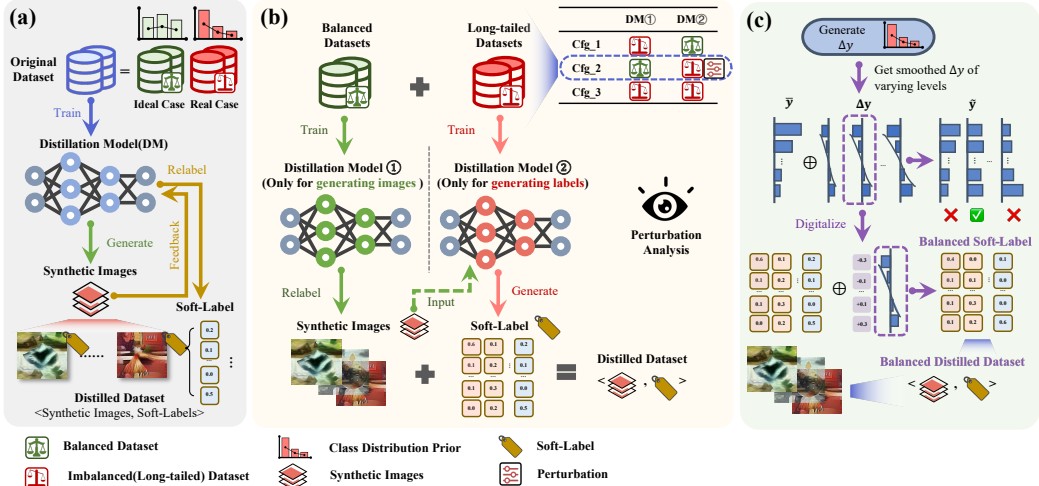

Figure 1: Overview of the experimental framework and modules. **(a)** The conventional dataset distillation pipeline utilizes a single model for both image and label generation. **(b)** The perturbation analysis framework employs two separate models: one for image synthesis and another for soft label generation. Three configurations indicate different combination of balanced/imbalanced dataset. We then perturb the imbalance levels to observe the resulting performance. **(c)** The proposed adaptive soft-label alignment module(ADSA). Symbols $\bar{y}$, $\Delta y$, and $\tilde{y}$ denote the average confidence, confidence adjustment by different levels, and the calibrated average confidence, respectively. We optimize $\Delta y$ to get the most uniform $\tilde{y}$ across classes, and use it to calibrate the soft labels. The operator $\oplus$ indicates addition in logit space; soft labels are shown here for visualization clarity.

each class contains the same number of samples as in the test set. While most existing works aim to improve distillation performance by focusing on insight (ii), such as EDC which enforces both global and class-aware feature matching corresponding to the second terms in Eq. 5 and Eq. 6, the widely used and empirically effective soft-labeling technique remains relatively underexplored. Notably, soft labels are highly generalizable and can be seamlessly integrated into various distillation pipelines. Therefore, this work primarily focuses on insight (i). In Section 3.2, we demonstrate that conventional soft-labeling strategies introduce bias in long-tailed dataset distillation, resulting in a mismatch in the first term of Eq. 5 and subsequent performance degradation. We further identify two main sources of this bias that are inherent in existing soft-labeling methods.

## 3.2 Perturbation Analysis and Dual Bias in Soft Labels

In this section, we first empirically validate that mismatches in soft labels (first term in Eq. 5) and distilled images (second term in Eq. 5) under long-tailed distributions lead to performance degradation. Motivated by Theorem 3.1, which underscores the importance of soft labels under imbalanced distributions, we further investigate the influence pathway from the original long-tailed data distribution to the resulting soft labels by identifying two key distortion factors. To investigate how imbalance affects predictions of the final evaluation model, we propose a perturbation analysis of distilled images and their corresponding soft labels, as illustrated in Figure 1. Unlike conventional dataset distillation pipelines where synthetic images and soft labels are jointly derived from the same distillation model, we decompose the process into two separate pipelines. This design enables controlled perturbation of either modality (distilled images or soft labels) by leveraging original datasets with varying degrees of class imbalance. By doing so, we can analyze the individual contribution of each modality to the downstream model's performance under long-tailed settings.

We employ an inversion-like objective [28]

$$\underset{D_{dd}}{\arg\min} \sum_{(\tilde{x}_{dd}, y) \in D_{dd}} L(f_{\theta_{tr}}(\tilde{x}_{dd}), y) + R_{reg}(\tilde{x}_{dd}), \tag{7}$$

to distill images, where $(\tilde{x}_{dd}, y) \in D_{dd}$, denotes a synthetic image-label pair, with $y$ as the predefined class label for each $\tilde{x}_{dd}$, and $R_{reg}(\tilde{x}_{dd})$ as the distribution matching regularization term such as feature distribution constraints $\|g(\tilde{x}_{dd}) - g(x_{ori})\|_2$ where $g$ denotes the network backbone of $f$. Soft labels are generated using the pretrained model. This follows the mainstream pipeline in recent large-scale dataset distillation frameworks [14, 16, 29, 30].

The original dataset, illustrated in the corresponding region of Figure 1, can be either balanced or imbalanced. The model trained on it is accordingly referred to as a balanced or imbalanced distillation model. We analyze four configurations[2] (referred to as configs for brevity) that differ in the composition of distillation models used to generate distilled images and assign soft labels: **config (1)** images distilled from an imbalanced model and labeled by a balanced model; **config (2)** images distilled from a balanced model and labeled by an imbalanced model; **config (3)** images both distilled and labeled by an imbalanced model; and **config (4)** images both distilled and labeled by a balanced model. Using the CIFAR-100 dataset [52], we designate 20 classes as tail classes and the remaining 80 as head classes. The number of images in tail classes is varied to simulate different levels of imbalance. The experimental details are provided in Appendix A.3. Figure 2 reports the confidence value (the predicted probabilities for the ground-truth class) for head and tail classes, the classification accuracy on tail classes, and the entropy of the softmax outputs. The x-axis represents the average number of samples across all tail classes or head classes. These metrics reflect the informativeness and reliability of the soft labels, both of which influence the evaluation model's performance, as discussed in [33].

As shown in Figure 2 (a), config (1) (using only imbalanced images), config (2) (using only an imbalanced distillation model for labeling), and config (3) (both imbalanced) all experienced a performance drop compared to config (4) (both balanced). This validates the theory presented in Section 3.1, which states that both biased distilled images and soft labels lead to performance degradation. Furthermore, we observed that config (2), which used an imbalanced model for labeling, showed a greater performance decrease than config (1). Figure 2 (b) illustrates the entropy of the soft labels, showing that a more imbalanced dataset leads to higher entropy, which indicates a lack of class-discriminative information in the soft labels [33].

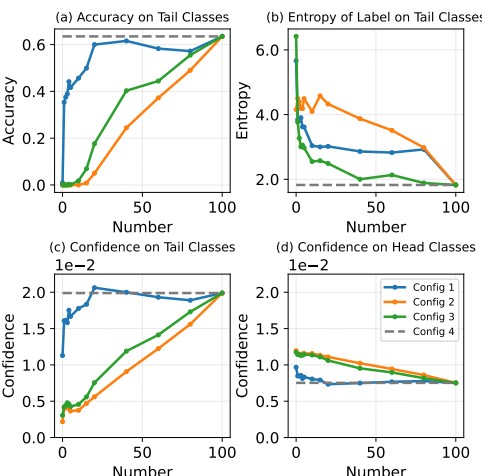

Figure 2: Effect of the number of images across head and tail classes on confidence, accuracy, and entropy. **(a)**: Accuracy trend on tail classes. **(b)**: Entropy of soft labels for tail classes. **(c)** and **(d)**: Confidence scores for tail and head classes with increasing tail samples.

To further analyze the respective contributions of imbalanced distilled images and imbalanced distillation models to the final soft labels, we visualize the per-class confidence. We use the term labeling model to refer to the distillation model used to generate soft labels for the distilled images. Panels (c) and (d) show that both imbalanced synthetic images and imbalanced labeling models lead to soft labels that are overconfident for head classes and underconfident for tail classes. Notably, even in config (1) where a balanced distillation model is used, its predictions are biased due to the imbalanced distilled images, and we refer to this as bias from distilled images. In contrast, the bias observed in config (2) stems from the prediction bias within the labeling model itself, which we term bias from the labeling model. As a result, the evaluation model trained on such biased soft labels from these two sources learns incorrect class probabilities, thereby inheriting this bias.

We can approximate the total bias in the soft labels using the following decomposition:

$$p_{DD}^{\text{obs}}(y \mid x) = p_{DD}^{\text{target}}(y \mid x) + \epsilon_T(y \mid x) + \epsilon_I(y \mid x), \tag{8}$$

where $p_{DD}^{\text{obs}}(y|x)$ denotes the observed soft label posterior, and $p_{DD}^{\text{target}}(y|x)$ represents the desired (unbiased) target posterior. The terms $\epsilon_T(y|x)$ and $\epsilon_I(y|x)$ correspond to the bias introduced by the labeling model and the synthetic images, respectively.

The above observation reveals the entangled sources of bias in soft labels and underscores the necessity of their calibration. In practice, however, the additive decomposition assumption may not hold, as evidenced in panel (a) of Figure 2, where the presence of both biases does not result in the worst performance. To address the entangled bias, we propose an effective lightweight calibration module that adaptively debiases the soft-label by utilizing the distilled images in next section.

---

[2]All networks $f$ involved in distillation will be trained separately for each configuration.

### 3.3 Adaptive Soft Label Alignment Module

The calibrated soft labels must satisfy three key properties. First, they should eliminate entangled biases to approximate the true posterior distribution on the test set, thereby tightening the generalization upper bound. Second, they should preserve the semantic relationships among classes to maintain the informative structure of the original soft labels, which is a key factor in enhancing the informational richness of the distilled dataset [33]. Third, the module should be adaptive across different datasets, image-per-class (IPC) regimes, and imbalance factor (IF) conditions to ensure robust generalization across varying scenarios.

The proposed module is illustrated in Figure 1(c). The main idea to address entangled biases is direct and unified: we adaptively calibrate the model's predictive outputs for each distilled image to obtain refined soft labels. To better illustrate the concept of ADSA, we present soft labels in Figure 1(c), although the calibration is actually applied to the logits.

First, to preserve semantic relationships among classes while smoothly adjusting each logit component, we adopt the logit calibration method proposed in [49]. The adjustment is defined as $\mathrm{argmax}_{y \in [K]} f_y(x) - \tau \cdot \log \pi_y$, where $\pi_y$ denotes the empirical frequency of class $y$ in the training set, $\tau$ is a calibration hyperparameter, $[K]$ is the label set, $f$ denotes the neural network, and $f_y(x)$ is the $y$-th logit output. While such methods are typically applied during inference to select the class with the highest logit and thus make a confident prediction, our focus lies in the relationship preserving property of soft labels. We obtain preliminary prediction probability for a given image:

$$p(y|x; \tau) = \frac{\exp(f_y(x) - \tau \log \pi_y)}{\sum_{y' \in [K]} \exp(f_{y'}(x) - \tau \log \pi_{y'})}. \tag{9}$$

We refer to $p(y|x)$ as the soft label without loss of generality.

We observe that the distilled images exhibit a distribution shift relative to the original training dataset. Consequently, the distilled set can serve as a hold-out validation set to diagnose class-wise output imbalance from the model trained on the original data. To quantify this, we compute the class-wise average soft label across all distilled images as $p(\bar{y} = i|x; \tau) = \mathbb{E}_{x \sim \mathcal{D}_i}[p(y = i \mid x; \tau)]$, where $\mathcal{D}_i$ denotes the set of distilled images labeled as class $i$. In Figure 1(c), the symbols $\bar{\mathbf{y}}$ and $\tilde{\mathbf{y}}$ correspond to $p(\bar{y}|x; 0)$ and $p(\bar{y}|x; \tau)$ respectively. To achieve balanced confidence across all classes, we optimize the calibration strength $\tau$ such that the class-wise confidence variance is minimized. The optimal $\tau^*$ is defined as

$$\tau^* = \mathrm{argmin}_{\tau} \sqrt{\frac{1}{K} \sum_{i=0}^{K-1} \left( p(\bar{y} = i|x; \tau) - \frac{1}{K} \sum_{j=0}^{K-1} p(\bar{y} = j \mid x; \tau) \right)^2}. \tag{10}$$

The soft labels are then calibrated as $p(y|x; \tau^*)$ and integrated into the final distilled dataset.

## 4 Experiment

In this section, we conduct a comprehensive evaluation of our dataset distillation method under long-tailed distribution settings. We begin by detailing the experimental setup, including datasets, evaluation metrics, and baseline methods for comparison. Specifically, we reproduce dataset distillation methods on long-tailed datasets and benchmark our approach against state-of-the-art baselines, demonstrating its superior performance under imbalanced conditions. We then perform ablation studies to assess the adaptiveness of our method, particularly in extreme long-tail scenarios, and evaluate its robustness under varying soft-label budgets. Finally, we provide explanatory analyses and visualizations to illustrate the effectiveness of the ADSA module.

**Experimental Setup.** We evaluate our method on CIFAR10/100-LT [52], and ImageNet-1k-LT ($224 \times 224$) [54]. The CIFAR-LT datasets are constructed using exponential long-tail distributions as in [42], with imbalance factor (IF) $r = \frac{n_0}{n_{K-1}}$ controlling class imbalance. Class sizes $n_i$ follow $n_i = n_0 \cdot r^{-\frac{i}{K-1}}$, and we test with $r \in 10, 50, 100$. The ImageNet-LT follows the setup in [43]. We evaluate performance across varying IPC (images per class) settings and compare our method with state-of-the-art baselines, including LTDD [11], SRe2L [14], GVBSM [30], and EDC [16].

Table 1: Comparison with baseline methods. Long-tailed datasets are constructed using an exponential decay in class frequency. Dataset distillation is then applied to generate compact synthetic datasets, on which evaluation models are evaluated. The table reports Top-1 validation accuracy on the distilled datasets. We use IF (imbalance factor) and IPC (images per class) to denote imbalance severity and distilled dataset size, respectively. LTDD [11] uses a depth-3 ConvNet as the backbone, SRe2L uses ResNet-18, and GVBSM/EDC adopt multiple architectures to distill and ResNet-18 to evaluate, as detailed in Appendix A.3.

| | CIFAR-10-LT | | | | | | | | |
| | IPC=1 | | | IPC=10 | | | IPC=50 | | |
| IF | 10 | 50 | 100 | 10 | 50 | 100 | 10 | 50 | 100 |
|---|---|---|---|---|---|---|---|---|---|
| LTDD | 28.0±1.0 | 24.4±0.3 | 23.8±0.5 | 58.1±0.3 | 54.2±1.0 | 53.4±0.1 | 70.5±0.4 | 65.8±0.2 | 64.0±0.9 |
| SRe2L | 16.9±2.2 | 17.8±2.3 | 14.3±1.7 | 24.1±0.7 | 23.3±1.9 | 22.6±0.4 | 40.4±0.6 | 36.6±1.9 | 34.6±1.4 |
| +ours | **20.2±0.9** | **18.1±1.2** | **19.3±1.3** | **27.1±1.6** | **25.0±0.2** | **25.9±0.6** | **47.1±0.1** | **38.8±0.6** | **45.3±0.7** |
| GVBSM | 34.2±0.4 | 28.9±0.4 | 25.1±0.3 | 49.5±0.1 | 33.8±0.3 | 29.4±0.1 | 58.2±0.2 | 37.2±0.0 | 30.9±0.0 |
| +ours | **39.1±0.4** | **36.0±0.6** | **31.5±0.5** | **54.4±0.5** | **45.8±0.1** | **40.4±0.4** | **64.7±0.0** | **51.4±0.2** | **46.9±0.2** |
| EDC | 32.3±1.4 | 29.5±0.8 | 30.0±0.6 | 69.9±0.7 | 58.5±0.5 | 50.9±0.5 | 77.8±0.1 | 65.6±0.3 | 56.0±0.2 |
| +ours | **35.2±0.8** | **36.4±1.0** | **39.3±0.6** | **73.2±0.4** | **69.8±0.2** | **68.7±0.3** | **80.8±0.2** | **76.4±0.4** | **74.8±0.1** |

| | CIFAR-100-LT | | | | | | | | |
| | IPC=1 | | | IPC=10 | | | IPC=50 | | |
| IF | 10 | 50 | 100 | 10 | 50 | 100 | 10 | 50 | 100 |
|---|---|---|---|---|---|---|---|---|---|
| LTDD | 12.3±0.2 | 11.1±0.1 | 10.6±0.1 | 31.5±0.2 | 26.8±0.3 | 24.9±0.1 | 40.0±0.1 | 34.5±0.1 | 31.6±0.0 |
| SRe2L | 7.4±0.3 | 8.5±0.6 | 7.5±0.1 | 27.0±1.4 | 25.0±2.5 | 22.6±1.1 | 41.4±1.8 | 32.8±2.0 | 28.9±0.4 |
| +ours | 6.4±0.0 | 7.1±0.4 | 7.0±0.8 | 25.9±0.5 | **26.3±0.2** | **23.2±0.1** | **47.2±0.3** | **42.5±0.9** | **37.6±2.8** |
| GVBSM | 20.3±0.3 | 19.7±0.1 | 19.1±0.2 | 35.1±0.0 | 31.2±0.2 | 27.9±0.1 | 39.2±0.3 | 34.0±0.1 | 31.1±0.1 |
| +ours | 20.0±0.3 | **20.4±0.3** | 17.0±0.6 | **35.5±0.1** | **32.8±0.1** | **29.6±0.3** | **40.1±0.0** | **35.8±0.3** | **33.2±0.4** |
| EDC | 42.3±0.0 | 34.3±0.3 | 32.0±0.1 | 54.3±0.6 | 43.4±1.6 | 39.0±1.8 | 57.0±0.8 | 45.7±1.6 | 40.9±1.8 |
| +ours | **43.4±0.4** | **37.9±0.5** | **34.1±0.3** | **55.5±0.1** | **46.0±0.1** | **41.6±0.1** | **58.0±0.2** | **48.1±0.1** | **43.5±0.2** |

Table 2: Evaluated on ImageNet-LT. Classes are categorized into head, medium, and tail using thresholds of 100 and 20 samples per class, respectively.

| Method | ImageNet-LT(top-1) | | | |
| | Head | Mid | Tail | Overall |
|---|---|---|---|---|
| SRe2L+IPC10 | 38.0 | 20.6 | 7.6 | 25.6 |
| +ours | 35.1 | **24.3** | **14.2** | **27.2** |
| SRe2L+IPC50 | 51.5 | 28.4 | 9.6 | 34.9 |
| +ours | 39.4 | **32.7** | **16.1** | **37.0** |
| SRe2L+IPC100 | 53.8 | 29.7 | 9.5 | 36.4 |
| +ours | 51.8 | **34.4** | **16.1** | **38.7** |
| EDC+IPC10 | 51.1 | 28.7 | 10.7 | 35.0 |
| +ours | 47.0 | **33.8** | **21.3** | **37.3** |
| EDC+IPC50 | 55.5 | 32.3 | 12.4 | 38.6 |
| +ours | 51.3 | **38.1** | **24.2** | **41.4** |
| EDC+IPC100 | 55.8 | 32.9 | 12.7 | 39.6 |
| +ours | 53.1 | **38.0** | **22.8** | **42.4** |

Table 3: Integration with MTT and DREAM on CIFAR-10-LT. * indicates results reported from LTDD [11].

| Method | IPC=10 | IPC=50 |
|---|---|---|
| *Random** | 33.2 | 51.6 |
| *DC** [19] | 37.3 | 35.8 |
| *IDM** [53] | 51.9 | 56.1 |
| *DATM** [24] | 41.6 | 50.3 |
| *LTDD** [11] | 54.2 | 65.8 |
| MTT [23] | 33.4 | 53.0 |
| + soft label | 37.9 | 51.4 |
| **+ ours** | **40.4** | **56.6** |
| DREAM [27] | 56.0 | 58.6 |
| + soft label | 41.0 | 46.7 |
| **+ ours** | **59.9** | **65.7** |
| EDC [16] | 77.9 | 64.4 |
| **+ ours** | **82.3** | **77.0** |

Table 4: Results under varying soft-label budgets. EP-$k$ denotes that soft labels are generated only for the first $k$ epochs then repeatedly used.

| EP | SRe2L | +Ours |
|---|---|---|
| 1 | 17.3 | 17.7(+0.4) |
| 10 | 32.8 | 35.1(+2.3) |
| 100 | 34.8 | 36.8(+2.0) |
| 300 | 34.9 | 37.0(+2.1) |

| EP | EDC | +Ours |
|---|---|---|
| 1 | 28.0 | 31.0(+3.0) |
| 10 | 37.3 | 40.0(+2.7) |
| 100 | 38.8 | 41.6(+2.8) |
| 300 | 38.7 | 41.9(+3.2) |

ResNet-18 is used as the default evaluation backbone for SRe2L, GVBSM, and EDC, while ConvNet is adopted for LTDD. For a fair comparison, we follow the original training hyperparameters for all methods, and report the top-1 accuracy.

## 4.1 Main Results

As shown in Table 1, our method consistently improves the performance of existing dataset distillation approaches across various IPC and imbalance factor (IF) settings. As IPC and IF increase, our method achieves further performance gains. Notably, integrating our method with SRe2L leads to improvements up to ~10% in overall accuracy on both CIFAR-10-LT and CIFAR-100-LT. Results on ImageNet-1k-LT (Table 2) further validate its effectiveness at scale. For example, with SRe2L at IPC=10, our method improves tail-class accuracy from 7.6% to 14.2% and overall accuracy from 25.6% to 27.2%. Similar trends are observed across all IPC settings and baselines. These gains highlight the robustness of our method in improving both average and tail-class performance for large-scale long-tailed dataset distillation. We further include in Appendix A.5 a comparison with enhanced baselines, where the distillation models are trained using resampling techniques.

Figure 3 (left) shows the class-wise accuracy. Without calibration, the model over-predicts head classes, causing frequent tail-class misclassification and reduced accuracy. In contrast, our method boosts both tail-class and overall performance. We also compare our method with classic long-tailed

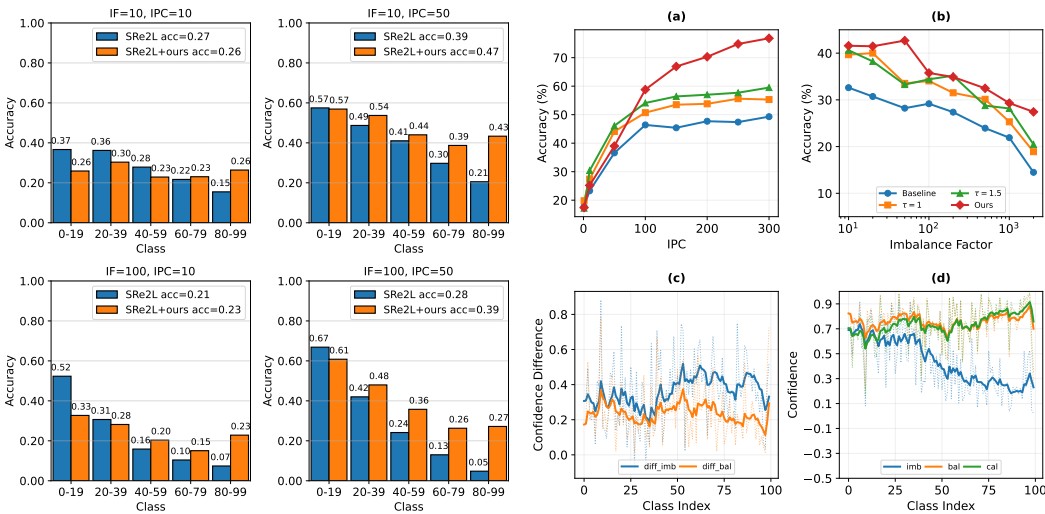

Figure 3: **Left:** Class-wise accuracy under different imbalance factors (IF) and images per class (IPC). Our method consistently improves overall and tail-class performance. **Right:** (a) Accuracy under varying IPC; (b) Accuracy under varying IF; (c) Class-wise confidence difference between original and distilled images; (d) Per-class soft-label confidence distributions. In panel (c) and (d)"imb"/"bal" denote soft labels from models trained on imbalanced/balanced original data; "cal" indicates calibrated soft labels 'imb' from the imbalanced model. Dashed lines represent raw outputs; solid lines are EMA-smoothed results.

recognition methods in Table 16, 17 (Appendix A.6), which shows that our approach provides a novel data-centric solution by directly transforming imbalanced data into a balanced and compact form, instead of modifying model architectures or loss functions.

## 4.2 Ablation Study

We first evaluate the effectiveness of our method across alternative dataset distillation frameworks and different soft-label budget settings. We then demonstrate the method's effectiveness in mitigating two types of bias and further assess the robustness of our adaptive strategy under extreme IPC and IF conditions. To more accurately reflect the performance of ADSA under varying imbalance factors (IF), we adopt different pretraining epochs for models corresponding to each IF, and report the optimal performance achieved by ADSA as used in the bias mitigation and extreme-IF evaluation experiments. More detailed results and analyses are provided in Appendix A.4.

To demonstrate the versatility of ADSA, we apply it to two representative dataset distillation approaches: MTT [23] and DREAM [27], which are based on trajectory matching and gradient matching, respectively. Unlike in SRe2L, where ADSA is applied for multiple epochs, here we integrate the ADSA module to generate calibrated soft labels using only a single distillation epoch. Table 3 summarizes the results under IPC = 10 and 50. Our method consistently improves performance across both MTT and DREAM, outperforming the original methods as well as their soft-label-enhanced variants. The improvements are especially notable on DREAM, where applying ADSA significantly boosts accuracy while preserving the simplicity of the original distillation framework.

Table 4 presents results under varying soft-label budgets. Conventional methods generate soft labels for each of the $N$ distilled images across all $K$ student training epochs, resulting in a total budget of $N \times K$ labels. Our method demonstrates its efficiency by operating on a smaller budget: we generate soft labels only for the first $K'$ epochs ($K' \leq K$) and then reuse this static set of labels for the full K-epoch training schedule. Though reducing the number of soft-labeling epochs in SRe2L

Table 5: Handling bias from soft labels. Y/N indicate presence/absence of bias.

| IF | biased img | biased label | method | IPC=1 | IPC=10 | IPC=50 |
|---|---|---|---|---|---|---|
| 10 | N | Y | SRe2L | 16.6 | 34.3 | 50.6 |
| | N | Y | +ours | **19.0** | **41.4** | **62.6** |
| 100 | N | Y | SRe2L | 18.3 | 24.4 | 31.5 |
| | N | Y | +ours | **21.8** | **36.9** | **57.6** |

Table 6: Handling bias from distilled images.

| IF | biased img | biased label | method | IPC=1 | IPC=10 | IPC=50 |
|---|---|---|---|---|---|---|
| 10 | Y | N | SRe2L | 16.0 | 34.7 | 55.3 |
| | Y | Y | +ours | **23.7** | **40.2** | **63.1** |
| 100 | Y | N | SRe2L | 15.6 | 22.9 | 54.8 |
| | Y | Y | +ours | **21.4** | **33.2** | **58.4** |

and EDC on ImageNet-1k-LT with IPC=50, our method consistently outperforms baselines, showing robustness under limited supervision.

We then verify the effectiveness of ADSA in mitigating the two types of biases discussed in Section 3.2, including the bias from distilled images and the bias from the distillation model. Table 5 shows that when the distilled images are obtained from an unbiased distillation model (training on balanced dataset) while the soft labels are derived from a biased one (training on imbalanced dataset), our method improves performance. Table 6 indicates that even when both the distilled images and soft labels are biased, our method still outperforms the baseline that uses soft labels from an unbiased distillation model. These two experiments verify that our method can effectively mitigate the two types of biases present in the soft labels.

To further assess the adaptiveness of the ADSA module, we examine its performance under varying IPC and imbalance factor (IF) settings, as shown in panels (a) and (b) in the right part of Figure 3. Results show that no single fixed calibration strength ($\tau$) consistently yields optimal performance across all settings, while ADSA achieves the best accuracy in most cases. The advantage is especially notable under extreme IPC and IF values. Being hyperparameter-free, ADSA dynamically adjusts calibration without tuning, ensuring robust and stable performance across scenarios.

### 4.3 Interpretability

Panel (c) in Figure 3 illustrates class-wise confidence differences, computed as the model's confidence on original training images minus that on distilled images. The resulting pattern reveals a domain shift between the two sources, particularly pronounced in tail classes, suggesting that distilled images can effectively act as a proxy hold-out set for calibration. Panel (d) in Figure 3 shows the per-class confidence distribution from different sources of soft labels. It shows that the soft labels from the ADSA module align well with the soft labels from the distillation model trained on a balanced dataset. More analysis and visualization are provided in Appendix A.7.

## 5 Conclusion

This paper is the first to examine the crucial role of soft labels in long-tailed dataset distillation. We propose an imbalance-aware generalization bound and conduct a perturbation analysis of existing distillation techniques. Our analysis reveals two sources of bias in soft labels, and we introduce a simple yet effective module that consistently improves performance over baselines across various settings, without requiring additional architectural complexity or computational overhead.

**Limitations and Future Work**  Our method assumes a shared conditional distribution between training and test sets, i.e., $p_{\text{tr}}(x|y) = p_{\text{te}}(x|y)$, which is common in long-tailed learning literature. However, this assumption may not hold in the presence of domain shift. Additionally, the current design relies on access to long-tail prior statistics and is not directly suited for online or streaming learning settings where such information is unavailable. Future work could focus on developing frequency-agnostic methods for online scenarios and exploring more efficient techniques for soft-label representation, storage, and utilization. Additionally, tailored feature-distribution matching strategies, such as transfer learning from head-to-tail classes and feature augmentation methods common in long-tailed recognition, may further enhance tail-class feature alignment.

## Acknowledgments

This work was supported by National Natural Science Foundation of China (grant No. 62350710797). We would like to thank the reviewers for their invaluable feedback and constructive comments to improve the paper.

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

# A Technical Appendices and Supplementary Material

## A.1 List of Notations

Table 7: Summary of frequently used notations.

| Symbol | Description |
|---|---|
| $D_{tr}$ | Training or original dataset |
| $D_{dd}$ | Distilled dataset |
| $D_{te}$ | Test dataset |
| $D$ | Dataset |
| $|D|$ | Size of dataset $D$ |
| $f_\theta$ | Model with parameters $\theta$ |
| $x$ | Input data (e.g., image) |
| $y$ | Label (e.g., one-hot label or soft label) |
| $l(\theta, D)$ | Loss function on dataset $D$ with model parameters $\theta$ |
| $L(f_\theta(x), y)$ | Cross-entropy loss on $(x, y)$ |
| $p_{tr}(x, y)$ | Joint distribution of training data and labels |
| $p_{dd}(x, y)$ | Joint distribution of distilled data and labels |
| $p_{te}(x, y)$ | Joint distribution of test data and labels |
| $\hat{p}(y|x)$ | Predicted probability of label $y$ given input $x$. Could refer to a model trained on $p_{dd}$ |
| $C$ | Upper bound of $-\log \hat{p}(y|x)$ |
| $R_{dd}$ | Regularization term in generalization bound in Eq. 4 |
| $n_k$ | Number of samples in class $k$ in $D_{tr}$ |
| $K$ | Number of classes |
| $D_{KL}(p\|q)$ | Kullback-Leibler divergence between distributions $p$ and $q$ |
| $const$ | Constant term independent of $p_{dd}(x, y)$ |
| $\tilde{x}_{dd}$ | Distilled images in the distillation process |
| $x_{ori}$ | Original images from the training dataset |
| $R_{reg}$ | Regularization term for distribution of distilled images and original images |
| $g(x)$ | Feature extractor function, the backbone of the model $f$ |
| $p_{DD}^{\text{obs}}(y|x)$ | The observed soft label posterior distribution |
| $p_{DD}^{\text{target}}(y|x)$ | The ideal or desired soft label posterior distribution(unbiased) |
| $\epsilon_T(y|x)$ | The bias introduced by the distillation model |
| $\epsilon_I(y|x)$ | The bias introduced by the distilled images |
| $[K]$ | The label set, i.e., $\{0, 1, \ldots, K-1\}$ |
| $f_y(x)$ | The $y$-th output of the model $f(x)$ |
| $\tau$ | Calibration hyperparameter |
| $\pi_y$ | Empirical frequency of class $y$ in $D_{tr}$ |

## A.2 Derivation of Theorem 3.1

In conventional dataset distillation, assuming the cross-entropy loss function:

$$l_{tr} = \mathbb{E}_{p_{tr}}[-\log \hat{p}(y|x)] \tag{11}$$

$$l_{dd} = \mathbb{E}_{p_{dd}}[-\log \hat{p}(y|x)] \tag{12}$$

where $D_{tr}$ represents the training dataset distribution, and $D_{dd}$ represents the distilled dataset distribution. Then, following [29], we have:

$$l_{tr} \leq l_{dd} + \frac{C}{2\sqrt{2}}\sqrt{D_{KL}\left(p_{tr}(x,y)\|p_{dd}(x,y)\right)} \tag{13}$$

which further leads to:

$$l_{te} \approx l_{tr} \leq l_{dd} + \frac{C}{2\sqrt{2}}\sqrt{D_{KL}\left(p_{tr}(x)\|p_{dd}(x)\right) + D_{KL}\left(p_{tr}(y\mid x)\|p_{dd}(y\mid x)\right)}. \tag{14}$$

The approximation $l_{tr} \approx l_{te}$ is justified by VC theory [50] for a sufficiently large training set, provided that the training and test distributions are identical ($p_{tr}(x,y) \sim p_{te}(x,y)$). Consequently, the upper bound on $l_{tr}$ can be applied to $l_{te}$, establishing the minimization of both the distilled loss $l_{dd}$ and the distributional discrepancy as a viable strategy for constraining the test loss.

However, in long-tailed scenarios, the assumption $p_{tr}(x,y) \sim p_{te}(x,y)$ does not hold. Therefore, we introduce $p_{te}$ while considering the conventional long-tail assumption $p_{tr}(x|y) = p_{te}(x|y)$ [51]:

$$p_{te}(x|y) = p_{tr}(x|y) \tag{15}$$

$$\Rightarrow p_{te}(x,y) = \frac{p_{te}(y)}{p_{tr}(y)}p_{tr}(x,y) \tag{16}$$

The conditional distribution assumption in long-tailed recognition, $p_{tr}(x|y) = p_{te}(x|y)$, remains valid in long-tailed dataset distillation. In this setting, the original training set is a long-tailed dataset $D_{tr}$, from which a distilled dataset $D_{dd}$ is derived. The model is trained on $D_{dd}$ and evaluated on a balanced test set $D_{te}$. Since $D_{tr}$ and $D_{te}$ are identical to those used in standard long-tailed learning, the long-tailed assumption naturally extends to the distillation scenario.

Substituting Eq. 16 into Eq. 13, we obtain:

$$l_{te} \leq l_{dd} + \frac{C}{2\sqrt{2}}\sqrt{D_{KL}\left(p_{te}(x,y)\|p_{dd}(x,y)\right)} \tag{17}$$

$$= l_{dd} + \frac{C}{2\sqrt{2}}\sqrt{D_{KL}\left(\frac{p_{te}(y)}{p_{tr}(y)}p_{tr}(x,y)\|p_{dd}(x,y)\right)} \tag{18}$$

Expanding the KL divergence:

$$D_{KL}\left(\frac{p_{te}(y)}{p_{tr}(y)}p_{tr}(x,y)\|p_{dd}(x,y)\right) = \int \frac{p_{te}(y)}{p_{tr}(y)}p_{tr}(x,y)\log\frac{p_{te}(y)p_{tr}(x,y)}{p_{tr}(y)p_{dd}(x,y)}dxdy \tag{19}$$

There are two possible simplifications for Eq. 18. **Form 1**:

$$D_{KL}\left(\frac{p_{te}(y)}{p_{tr}(y)}p_{tr}(x,y)\|p_{dd}(x,y)\right) \tag{20}$$

$$= \int \frac{p_{te}(y)}{p_{tr}(y)}p_{tr}(x,y)\log\frac{p_{te}(y)p_{tr}(x,y)}{p_{tr}(y)p_{dd}(x,y)}dxdy \tag{21}$$

$$= \int \frac{p_{te}(y)}{p_{tr}(y)}p_{tr}(x,y)\log\frac{p_{tr}(y|x)}{p_{dd}(y|x)}dxdy + \int \frac{p_{te}(y)}{p_{tr}(y)}p_{tr}(x,y)\log\frac{p_{tr}(x)}{p_{dd}(x)}dxdy \tag{22}$$

$$+ \int \frac{p_{te}(y)}{p_{tr}(y)}p_{tr}(x,y)\log\frac{p_{te}(y)}{p_{tr}(y)}dxdy$$

$$= \int p_{te}(x,y)\log\frac{p_{tr}(y|x)}{p_{dd}(y|x)}dxdy + \int p_{te}(x,y)\log\frac{p_{tr}(x)}{p_{dd}(x)}dxdy \tag{23}$$

$$+ \int p_{te}(y)\log\frac{p_{te}(y)}{p_{tr}(y)}dxdy \tag{24}$$

$$= \int p_{te}(x,y)\log\left(\frac{p_{tr}(y|x)}{p_{te}(y|x)}\frac{p_{te}(y|x)}{p_{dd}(y|x)}\right)dxdy + \int p_{te}(x,y)\log\left(\frac{p_{tr}(x)}{p_{te}(x)}\frac{p_{te}(x)}{p_{dd}(x)}\right)dxdy \tag{25}$$

$$+ D_{KL}(p_{te(y)}\|p_{tr(y)})$$

$$= D_{KL}(p_{te}(y|x)\|p_{dd}(y|x)) - D_{KL}(p_{te}(y|x)\|p_{tr}(y|x)) \tag{26}$$

$$+ D_{KL}(p_{te}(x)\|p_{dd}(x)) - D_{KL}(p_{te}(x)\|p_{tr}(x)) + D_{KL}(p_{te}(y)\|p_{tr}(y))$$

$$= D_{KL}(p_{te}(y|x)\|p_{dd}(y|x)) + D_{KL}(p_{te}(x)\|p_{dd}(x)) + \text{const} \tag{27}$$

$$= D_{KL}(p_{te}(y|x)\|p_{dd}(y|x)) + D_{KL}\left(\int p_{te}(x|y)p_{te}(y)dy \middle\| \int p_{dd}(x|y)p_{dd}(y)dy\right) + \text{const} \tag{28}$$

Here, the const term groups all expressions that are independent of our optimization target $p_{dd}(x, y)$.

**Form 2**:

$$D_{KL}\left(\frac{p_{te}(y)}{p_{tr}(y)}p_{tr}(x,y)\|p_{dd}(x,y)\right) \tag{29}$$

$$= \int \frac{p_{te}(y)}{p_{tr}(y)}p_{tr}(x,y)\log\frac{p_{te}(y)p_{tr}(x,y)}{p_{tr}(y)p_{dd}(x,y)}dxdy \tag{30}$$

$$= \int \frac{p_{te}(y)}{p_{tr}(y)}p_{tr}(x,y)\log\frac{p_{tr}(x|y)}{p_{dd}(x|y)}dxdy + \int \frac{p_{te}(y)}{p_{tr}(y)}p_{tr}(x,y)\log\frac{p_{te}(y)p_{tr}(y)}{p_{tr}(y)p_{dd}(y)}dxdy \tag{31}$$

$$= \int p_{te}(y)p_{tr}(x|y)\log\frac{p_{tr}(x|y)}{p_{dd}(x|y)}dxdy + \int p_{te}(y)\log\frac{p_{te}(y)p_{tr}(y)}{p_{tr}(y)p_{dd}(y)}dy \tag{32}$$

$$= \int p_{te}(y)D_{KL}\left(p_{tr}(x|y)\|p_{dd}(x|y)\right)dy + D_{KL}(p_{te}(y)\|p_{dd}(y)) \tag{33}$$

Thus, two possible upper bounds exist: **Form 1**:

$$l_{te} = l_{dd} + \frac{C}{2\sqrt{2}}\sqrt{D_{KL}(p_{te}(y|x)\|p_{dd}(y|x)) + D_{KL}(p_{te}(x)\|p_{dd}(x)) + const} \tag{34}$$

$$= l_{dd} + \frac{C}{2\sqrt{2}}\sqrt{R_{dd}} \tag{35}$$

$$R_{dd} = D_{KL}(p_{te}(y|x)\|p_{dd}(y|x)) + D_{KL}\left(\int p_{te}(x|y)p_{te}(y)dy\middle\|\int p_{dd}(x|y)p_{dd}(y)dy\right) + const \tag{36}$$

**Form 2**:

$$l_{te} = l_{dd} + \frac{C}{2\sqrt{2}}\sqrt{\int p_{te}(y)D_{KL}\left(p_{tr}(x|y)\|p_{dd}(x|y)\right)dy + D_{KL}(p_{te}(y)\|p_{dd}(y))} \tag{37}$$

In Eq. 36, the *first term* suggests that the classifier $p_{dd}(y|x)$ trained on the distilled dataset should align with the classifier $p_{te}(y|x)$ obtained from long-tail calibration. This means that any long-tail calibration model trained on $p_{tr}(x,y)$ can serve as a teacher model in dataset distillation. The *second term* indicates that the overall distribution of the distilled dataset should align with the test distribution, which can be estimated using methods like Gaussian mixture models to promote better alignment.

In Eq. 37, the *first term* suggests ensuring that the statistical information of each class remains consistent, meaning that the conditional distribution $p_{tr}(x|y)$ in the training set should align with the corresponding distribution $p_{dd}(x|y)$ in the distilled dataset. The *second term* indicates that the class distribution of the distilled dataset should match that of the test set, which implies maintaining a fixed number of instances per class (IPC).

Since we assume $p_{te}(y) = \frac{1}{K}$, we obtain:

$$D_{KL}\left(p_{tr}(x|y)\|p_{dd}(x|y)\right) = 0 \tag{38}$$

which satisfies both:

$$\int p_{te}(y)D_{KL}\left(p_{tr}(x|y)\|p_{dd}(x|y)\right)dy = 0 \tag{39}$$

and

$$D_{KL}\left(\int p_{te}(x|y)p_{te}(y)dy\middle\|\int p_{dd}(x|y)p_{dd}(y)dy\right) = 0 \tag{40}$$

### A.3 Experiment Details

**Distillation Pipeline** The baseline methods we adopt (e.g. SRe2L [14], GVBSM [30], and EDC [16]) follow the same dataset distillation pipeline, as illustrated in Fig. 1(a), which consists of three stages. The first stage (squeeze stage) involves training a distillation model on the original dataset. The second stage (recover stage) generates distilled images by optimizing the objective in Eq. 7. The third stage (relabel stage) uses the distillation model to predict soft labels for the distilled images. The resulting distilled images and their corresponding soft labels are then used together to evaluate the quality of the distilled dataset by measuring the performance of student models trained on it.

**Hyperparameters** For all baselines, we adopt their default hyperparameter settings, optimizer and augmentation strategy except for special demonstration. The optimization problem in Equation 10 is solved by performing a search over the range $\tau \in (0, 3)$. Specifically, DREAM splits each distilled image into four individual clips and resizes them to train the evaluation model. We generate a soft label for each clip once per epoch. In the soft label budget experiment, we generate soft labels for $k$ epochs and use only these $k$-epoch soft labels to train the evaluation model for 300 epochs. Since part of the data in LTDD [11] is missing, we manually reproduced the results for CIFAR-10 with IPC=1, for CIFAR-100 with IPC=1 across all imbalance factors (IF), and for IPC=10 and 50 under IF=50 and 100. For all other settings in LTDD, we used the data reported in their paper.

**Architectures** We use a depth-3 ConvNet as both the evaluation and distillation backbone for MTT and DREAM, and adopt ResNet-18 as the evaluation backbone for SRe2L, GVBSM, and EDC. All backbones are manually trained on long-tailed CIFAR and ImageNet-1k datasets without using any pretrained models from the PyTorch model zoo. The backbone used is shown in Table 8.

For CIFAR dataset distillation, we use the default model architectures provided in the official codebases. For ImageNet-1k, the original GVBSM and EDC papers employed EfficientNet-B0 and ShuffleNet-V2-x0.5, respectively. Due to the lack of publicly available training code for these models on ImageNet-1k, we replace them with EfficientNet-V2-S and ShuffleNet-V2-x1.5. We use the mean of calibrated soft label from trained distillation models as the final soft label.

**Compute Resources** All experiments are conducted on an 8 RTX 4090 GPUs server, and the computational cost depends on the underlying baseline methods. When IPC = 50, dataset distillation on CIFAR-10/CIFAR-100 typically takes less than 10 GPU hours. Among the methods, SRe2L is the fastest (around 2 GPU hours), while GVBSM is the slowest (around 10 GPU hours). On ImageNet, the SRe2L is the fastest (around 2 GPU days) and GVBSM the slowest (around 15 GPU days). The exact runtime may fluctuate depending on the GPU model, GPU/CPU utilization, and inter-GPU communication efficiency. The additional computational and memory overhead introduced by ADSA is negligible.

**Perturbation Analysis Details** The perturbation study was performed on CIFAR100 with 100 classes, we limit the total number of images to be 10000 images, but the number of images for a head class could exceed 100, and then we distribute $a$ images for the first 20 classes, then we distribute $(10000 - 20 * a)/(100 - 20)$, and then round to integer. We use 10000 images to avoid the number of head class to exceed the maximum number(500) for each class.

The perturbation study is conducted on CIFAR-100, which contains 100 classes. We limit the total number of images to 10,000 to ensure that the number of images per head class does not exceed the dataset's upper limit of 500 images per class. Specifically, we assign $a$ images to each of the first 20 classes (head classes), and distribute the remaining images uniformly across the remaining 80 classes (tail classes) using $\frac{10,000 - 20 \cdot a}{100 - 20}$, followed by rounding to the nearest integer. Fixing the total number of images, rather than using an exponentially decaying class frequency distribution, allows us to control the dataset size and eliminate its influence on performance. This enables a more controlled evaluation of the impact of tail-class sample sizes. Furthermore, by averaging the statistics over the first 20 classes, we can quantitatively assess performance across varying degrees of class imbalance while minimizing the influence of individual class identities.

### A.4 Analysis of Pretraining Epoch Selection

In Tables 9 and 10, we present the performance of distilled datasets generated by distillation models trained with different pretraining epochs on CIFAR-10 and CIFAR-100. Tables 11 and 12 summarize the best accuracy achieved under each imbalance factor (IF) by selecting the optimal epoch, while Tables 13 and 14 report the corresponding optimal epochs. In these tables, each column corresponds to a different IPC setting: IPC = 1, 10, or 50. The results clearly indicate that the choice of training epochs has a substantial impact on the quality of the distilled dataset, with performance variations exceeding 10% in some cases. Moreover, we observe that the optimal number of epochs tends to increase with higher IF and IPC values. This suggests that for easier cases (e.g., low IF or IPC), early-stage models are sufficient to yield high-quality distilled data, whereas more challenging scenarios require prolonged training to achieve optimal performance. Notably, Tables 11 and 12 can be viewed as enhanced versions of the results in Table 1, where SRe2L benefits from epoch tuning. Under

Table 8: Backbone architectures used by SRe2L, GVBSM, and EDC on CIFAR-10, CIFAR-100. The models are used to distill images and assign soft label.

| Method | CIFAR10 | CIFAR100 | ImageNet-1k |
|---|---|---|---|
| SRe2L | ResNet-18 | ResNet-18 | ResNet-18 |
| GVBSM | ResNet-18, ConvNet-W128, MobileNetV2, WRN-16-2, ShuffleNet-V2-x0.5 | ResNet-18, ConvNet-W128, MobileNetV2, WRN-16-2, ShuffleNet-V2-x0.5 | ResNet-18, MobileNetV2, EfficientNetV2-S, ShuffleNet-V2-x0.5 |
| EDC | ResNet-18, ConvNet-W128, MobileNetV2, WRN-16-2, ShuffleNet-V2-x0.5, ConvNet-D1, ConvNet-D2, ConvNet-W32 | ResNet-18, ConvNet-W128, MobileNetV2, WRN-16-2, ShuffleNet-V2-x0.5, ConvNet-W32, ConvNet-D1, ConvNet-D2 | ResNet-18, MobileNetV2, EfficientNetV2-S, ShuffleNet-V2-x0.5, AlexNet |

this enhanced setting, the performance gap between our method and the baselines further widens, particularly under high IPC, highlighting the robustness and scalability of our approach.

We adopt this flexible epoch selection strategy in Figure 3(b), Table 5, Table 6, and Table 15 to more accurately evaluate and reflect the effectiveness of our method. In particular, for Figure 3(b), using a fixed number of training epochs (e.g., 200) to match the baseline setting may result in anomalous accuracy trends, specifically, performance may even increase with higher imbalance factors (IF), which contradicts expected behavior. This is because a fixed epoch value is not necessarily optimal, and the optimal epoch often shifts later as the imbalance increases.

Table 9: CIFAR-10 accuracy under different pre-training epochs

| Epoch | IF | Method | 1 | 10 | 50 |
|---|---|---|---|---|---|
| 10 | 10 | baseline | 18.0 | 39.6 | 55.0 |
| | | ours | 23.1 | 41.9 | 63.2 |
| | 50 | baseline | 20.4 | 22.8 | 24.6 |
| | | ours | 26.2 | 34.9 | 38.7 |
| | 100 | baseline | 16.3 | 21.6 | 23.6 |
| | | ours | 23.0 | 33.0 | 37.9 |
| 30 | 10 | baseline | 14.4 | 27.2 | 51.5 |
| | | ours | 16.7 | 26.2 | 50.0 |
| | 50 | baseline | 14.1 | 26.4 | 50.3 |
| | | ours | 18.2 | 30.7 | 63.7 |
| | 100 | baseline | 15.2 | 28.1 | 38.2 |
| | | ours | 22.0 | 32.0 | 59.1 |
| 50 | 10 | baseline | 14.2 | 24.6 | 46.8 |
| | | ours | 15.7 | 25.2 | 53.6 |
| | 50 | baseline | 15.4 | 26.5 | 39.9 |
| | | ours | 18.7 | 32.1 | 52.1 |
| | 100 | baseline | 17.9 | 23.9 | 40.3 |
| | | ours | 19.4 | 29.0 | 53.8 |
| 70 | 10 | baseline | 14.0 | 23.5 | 41.4 |
| | | ours | 17.0 | 25.8 | 50.8 |
| | 50 | baseline | 13.0 | 22.3 | 39.3 |
| | | ours | 14.3 | 29.0 | 56.2 |
| | 100 | baseline | 13.2 | 21.2 | 40.3 |
| | | ours | 15.3 | 28.1 | 54.7 |

Table 10: CIFAR-100 accuracy under different pretraining epochs

| Epoch | IF | Method | 1 | 10 | 50 |
|---|---|---|---|---|---|
| 10 | 10 | baseline | 17.3 | 31.7 | 32.6 |
| | | ours | 17.3 | 37.2 | 39.3 |
| | 50 | baseline | 17.5 | 21.5 | 21.9 |
| | | ours | 18.1 | 25.5 | 25.6 |
| | 100 | baseline | 15.7 | 17.5 | 17.6 |
| | | ours | 7.7 | 18.0 | 17.8 |
| 50 | 10 | baseline | 8.4 | 35.4 | 43.7 |
| | | ours | 7.0 | 35.1 | 53.3 |
| | 50 | baseline | 9.6 | 28.5 | 33.0 |
| | | ours | 8.9 | 34.2 | 43.3 |
| | 100 | baseline | 10.2 | 29.0 | 31.4 |
| | | ours | 9.3 | 33.0 | 39.2 |
| 100 | 10 | baseline | 9.1 | 34.9 | 46.1 |
| | | ours | 7.6 | 33.2 | 53.7 |
| | 50 | baseline | 8.2 | 30.1 | 35.0 |
| | | ours | 7.7 | 31.5 | 45.3 |
| | 100 | baseline | 8.9 | 28.1 | 31.8 |
| | | ours | 7.7 | 31.6 | 41.7 |
| 200 | 10 | baseline | 7.4 | 27.0 | 41.4 |
| | | ours | 6.4 | 25.9 | 47.2 |
| | 50 | baseline | 8.5 | 25.0 | 32.8 |
| | | ours | 7.1 | 26.3 | 42.5 |
| | 100 | baseline | 7.5 | 22.6 | 28.9 |
| | | ours | 7.0 | 23.2 | 37.6 |

### A.5 Comparison with Enhanced Resampling Baseline

Table 15 compares our method with a enhanced baseline, where distillation models are trained using a class-balanced resampling strategy that ensures each class appears with equal frequency during pre-training. The experimental results show that our method outperforms both the baseline and the

Table 11: CIFAR-10 max accuracy for different epochs

| IF | Method | 1 | 10 | 50 |
|---|---|---|---|---|
| 10 | baseline | 18.0 | 39.6 | 55.0 |
| | ours | 23.1 | 41.9 | 63.2 |
| 50 | baseline | 20.4 | 26.5 | 50.3 |
| | ours | 26.2 | 34.9 | 63.7 |
| 100 | baseline | 17.9 | 28.1 | 40.3 |
| | ours | 23.0 | 33.0 | 59.1 |

Table 12: CIFAR-100 max accuracy for different epochs

| IF | Method | 1 | 10 | 50 |
|---|---|---|---|---|
| 10 | baseline | 9.1 | 35.4 | 46.1 |
| | ours | 7.6 | 37.2 | 53.7 |
| 50 | baseline | 9.6 | 30.1 | 35.0 |
| | ours | 8.9 | 34.2 | 45.3 |
| 100 | baseline | 8.9 | 29.0 | 31.8 |
| | ours | 9.3 | 33.0 | 41.7 |

Table 13: CIFAR-10 best epoch

| IF | Method | 1 | 10 | 50 |
|---|---|---|---|---|
| 10 | baseline | 10 | 10 | 10 |
| | ours | 10 | 10 | 10 |
| 50 | baseline | 10 | 50 | 30 |
| | ours | 10 | 10 | 30 |
| 100 | baseline | 50 | 30 | 50 |
| | ours | 10 | 10 | 30 |

Table 14: CIFAR-100 best epoch

| IF | Method | 1 | 10 | 50 |
|---|---|---|---|---|
| 10 | baseline | 100 | 50 | 100 |
| | ours | 100 | 10 | 100 |
| 50 | baseline | 50 | 100 | 100 |
| | ours | 50 | 50 | 100 |
| 100 | baseline | 100 | 50 | 100 |
| | ours | 50 | 50 | 100 |

resample setting in most cases, while resampling indeed improves over the baseline. The results show that our method could be seamlessly integrated as a plug-and-play module to further boost the performance of resampling-based pipelines in most cases. It is important to note that, to enable a fair comparison with the resampling-based methods, we also adopted the best-epoch selection strategy for all configurations (SRe2L, +resample, +ours, +resample+ours). For each imbalance factor (IF), the number of training epochs was selected to ensure optimal performance. Specifically, epochs were selected from $\{10, 30, 50\}$ for CIFAR-10, and from $\{10, 50, 100, 200\}$ for CIFAR-100. The results are also slightly better than those in Table 1.

### A.6 Comparison with Classic Long-tailed Recognition Methods

We also compare the results of our method with classic long-tailed recognition approaches in Table 16 and Table 17(as reported in [10]) to demonstrate that our method provides a data-centric solution for transforming imbalanced data into a balanced form, enabling models to be trained directly without requiring specialized designs for long-tailed scenarios.

### A.7 Further Ablation and Visualization

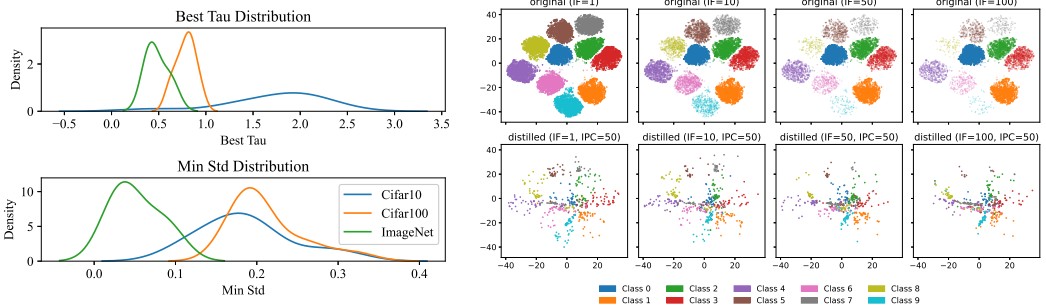

Figure 4: **Left:** Distribution of selected $\tau$ across different datasets. The value of $\tau$ varies within the same dataset due to differences in IPC, IF, and the random seed used in training. **Right:** t-SNE visualization of long-tailed and distilled datasets under different imbalance factors (IF). The top row shows the original long-tailed dataset distributions, which remain well-separated across different IF values. The bottom row presents the corresponding distilled datasets, where class clusters become increasingly compressed and less distinguishable as the imbalance factor increases.

Table 15: Performance comparison with a resampling-enhanced baseline on CIFAR-10/100-LT.

**CIFAR-10-LT**

| | IPC=1 | | | IPC=10 | | | IPC=50 | | |
|---|---|---|---|---|---|---|---|---|---|
| Imbalance Factor | 10 | 50 | 100 | 10 | 50 | 100 | 10 | 50 | 100 |
| SRe2L | 19.9 | 19.9 | 18.7 | 39.0 | 27.2 | 21.3 | 55.2 | 51.1 | 24.0 |
| +resample | 18.1 | 21.7 | **25.5** | 38.1 | 41.3 | 36.4 | 54.8 | 54.6 | 49.5 |
| +ours | **23.7** | **26.9** | 21.4 | **40.2** | 35.4 | 33.2 | **63.1** | **63.7** | **58.4** |
| +resample+ours | 19.5 | 22.3 | 24.1 | 38.0 | **44.1** | **37.1** | 58.8 | 56.6 | 56.1 |

**CIFAR-100-LT**

| | IPC=1 | | | IPC=10 | | | IPC=50 | | |
|---|---|---|---|---|---|---|---|---|---|
| Imbalance Factor | 10 | 50 | 100 | 10 | 50 | 100 | 10 | 50 | 100 |
| SRe2L | 16.3 | 17.8 | 15.9 | 35.8 | 28.8 | 29.1 | 45.2 | 33.5 | 31.8 |
| +resample | **20.3** | **18.9** | **16.3** | 35.8 | 29.2 | 28.4 | 45.1 | 36.1 | 32.8 |
| +ours | 16.9 | 18.2 | 9.6 | **38.2** | **34.2** | **33.0** | **53.9** | **45.6** | **41.7** |
| +resample+ours | 18.5 | 17.6 | 15.1 | 37.6 | 32.2 | 29.9 | 50.6 | 39.6 | 34.3 |

Table 16: Comparison with Long-Tailed Recognition Methods on CIFAR

| Method | CIFAR-10-LT(top-1) Imbalance Factor | | | CIFAR-100-LT(top-1) Imbalance Factor | | |
|---|---|---|---|---|---|---|
| | 100 | 50 | 10 | 100 | 50 | 10 |
| Softmax Loss[55] | 70.3 | 74.8 | 86.3 | 38.2 | 43.8 | 55.7 |
| Focal Loss[56] | 70.3 | 76.7 | 86.6 | 38.4 | 44.3 | 51.9 |
| CB Loss[46] | 74.5 | 79.2 | 87.4 | 39.6 | 45.3 | 57.9 |
| BBN[42] | 79.8 | 82.1 | 88.3 | 43.3 | 48.5 | 55.6 |
| TSC[57] | 79.7 | 82.9 | 88.7 | 43.8 | 47.4 | 59.0 |
| EDC10+ours | 68.8 | 72.1 | 76.3 | 41.8 | 46.0 | 55.5 |
| EDC50+ours | 74.0 | 77.0 | 82.3 | 43.8 | 47.9 | 58.1 |

Table 17: Comparison with Long-Tailed Recognition Methods on ImageNet

| Method | Backbone | ImageNet-LT(top-1) | | | |
|---|---|---|---|---|---|
| | | Head | Mid | Tail | Overall |
| Softmax loss[55] | ResNet-10 | 40.9 | 10.7 | 0.4 | 20.9 |
| Focal loss[56] | ResNet-10 | 36.4 | 29.9 | 16.0 | 30.5 |
| Range loss[58] | ResNet-10 | 35.8 | 30.3 | 17.6 | 30.7 |
| OLTR[43] | ResNet-10 | 43.2 | 35.1 | 18.5 | 35.6 |
| FSA[44] | ResNet-10 | 47.3 | 31.6 | 14.7 | 35.2 |
| cRT[48] | ResNeXt-50 | - | - | - | 39.5 |
| SRe2L+ours | ResNet-18 | 35.1 | 24.3 | 14.2 | 27.2 |
| SRe2L+ours | ResNet-18 | 39.4 | 32.7 | 16.1 | 37.0 |
| SRe2L+ours | ResNet-18 | 51.8 | 34.4 | 16.1 | 38.7 |
| EDC+ours | ResNet-18 | 47.0 | 33.8 | 21.3 | 37.3 |
| EDC+ours | ResNet-18 | 51.3 | 38.1 | 24.2 | 41.4 |
| EDC+ours | ResNet-18 | 53.1 | 38.0 | 22.8 | 42.4 |

We visualize the distribution of the optimal calibration parameter $\tau$ across different datasets in Figure 4 left. All subfigures use the same feature extractor, which is the backbone of a ResNet-18 model trained for 200 epochs on the balanced original dataset. We pass all images including both the original and the distilled images under different imbalance factors (IFs) through this feature extractor and jointly project them into a low-dimensional space using t-SNE. This ensures that their spatial positions are directly comparable in the visualization. Figure 4 demonstrates that the optimal calibration strength varies significantly with the dataset. Additionally, Figure 4 right presents a t-SNE visualization of the original and distilled image distributions. While the original datasets maintain a relatively uniform distribution in the feature space, regardless of the imbalance factor, the distribution of the distilled images becomes increasingly compressed and distorted as the imbalance factor increases. At lower imbalance factors, the distilled samples are more evenly dispersed across the feature space. However, as the imbalance factor grows, these samples concentrate in a smaller subregion of the space, deviating substantially from the distribution of the original dataset. This distortion may explain why a model trained on the distilled dataset, despite being balanced, can still produce biased soft labels, as the underlying feature representation remains inherently skewed.

