# OpenReview forum: "Rectifying Soft-Label Entangled Bias in Long-Tailed Dataset Distillation"
_NeurIPS.cc/2025/Conference — NeurIPS 2025 poster_

### Official Review · Reviewer_cicZ · 2025-06-21

**Clarity:** 3
**Significance:** 4
**Originality:** 4
**Rating:** 5
**Confidence:** 3

**Summary:**

This paper addresses the important problem of distilling or compressing real-world long-tail data, which is highly relevant in many practical scenarios. A particularly notable contribution is the derivation of an imbalance-aware generalization bound for models trained on distilled datasets, providing valuable theoretical insight. Additionally, the proposed ADSA module is a key strength of the work, offering seamless integration into existing distillation pipelines, which enhances its practical applicability.

**Questions:**

**Major**
- ADSA performs soft label calibration to address the bias introduced by the distillation model. However, it is unclear whether this also mitigates the bias inherent in the distilled images themselves. I believe the paper would benefit from an explicit explanation of how ADSA handles this aspect.
- In Figure 3(b), why is there a sudden performance improvement at the end, even as the imbalance factor increases? What explains this behavior?
- In scenarios with small IPC, the performance of ADSA sometimes drops. Do the authors have any insight or explanation for this?
- In Figure 1(b), the balanced datasets trained model is stated to be used only for relabeling, yet it appears to be generating images. Similarly, the model used only for generating images also seems to produce labels. Could the authors clarify whether this is a typo or an inconsistency in the figure?

**Minor**
- When applied to a balanced training dataset, does ADSA lead to any performance degradation?
- It seems that additional computation is required for ADSA. As the number of data points and classes increases, does the distillation speed or memory usage become a bottleneck?

**Ethical Concerns:**

["NO or VERY MINOR ethics concerns only"]

**Final Justification:**

After carefully revisiting both the authors’ rebuttal and the main paper, I acknowledge that the authors have addressed most of my concerns—particularly those related to bias in distilled images and certain empirical inconsistencies in the main paper. I also appreciate the clarification regarding the applicability to balanced datasets and the potential computational or memory bottlenecks as the number of data points or classes increases.

**Limitations:**

yes

**Paper Formatting Concerns:**

No concerns.

**Quality:**

4

**Strengths And Weaknesses:**

**Strengths**
- While I am not an expert in dataset distillation, the paper clearly communicates its core ideas and the importance of the problem it addresses. It is a well-written and accessible paper even for readers less familiar with the domain.
- The experimental evaluation is thorough, and the inclusion of diverse visualization results is particularly impressive.
- The method demonstrates consistent performance improvements across a wide range of baselines, which highlights its robustness.
- The proposed hyperparameter-free approach is a significant advantage, as it enhances the method’s practicality and ease of adoption in real-world scenarios.

---

> ### Author Rebuttal · Authors · 2025-07-31
>
> We sincerely thank the reviewer for the positive evaluation and insightful suggestions. We have carefully addressed all comments through additional experiments and clarification below.
>
> **Q1: It is unclear whether ADSA module also mitigates the bias inherent in the distilled images themselves.**
>
> **A:** Our proposed method could mitigate the two bias sources (distilled images and distillation model) in soft label.  A key contribution of our method is its ability to implicitly rectify both sources of bias on soft labels including distilled image bias and soft-label bias in a unified manner. We conducted two additional experiments to prove that our method could mitigate both the bias succesfully.
>
> Exp1: Addressing bias from the distillation model (biased soft labels)
>
> We use soft labels produced by a model trained on an long-tailed dataset, while distilled images are obtained from a model trained on a balanced dataset.
>
> | IF | biased img | biased label | method | IPC=1 | IPC=10 | IPC=50 |
> | - | - | - | - | - | - | - |
> | 10   | N  | Y | SRe2L  | 16.6     | 34.3     | 50.6     |
> |    | N | Y | +ours  | **19.0** | **41.4** | **62.6** |
> | 100  | N | Y  | SRe2L  | 18.3     | 24.4     | 31.5     |
> |   | N | Y | +ours  | **21.8** | **36.9** | **57.6** |
>
> > Note: "Y" indicates the presence of bias; "N" indicates no bias in that source. "IF" means Imbalance Factor.
>
> These results clearly show that our method can effectively mitigate soft-label bias introduced by a biased distillation model.
>
> Exp2: Addressing bias in distilled images
>
> This experiment evaluates the bias introduced by distilled images. The baseline uses images from a long-tailed model and soft labels from a balanced one, while our method is applied when both images and labels are from long-tailed models.
>
> | IF   | biased img | biased label | method | IPC=1 | IPC=10 | IPC=50 |
> | - | - | - | - | - | - | - |
> | 10   | Y  | N | SRe2L  | 16.0 | 34.7 | 55.3  |
> |      | Y | Y  | +ours  | **23.7** | **40.2** | **63.1** |
> | 100  | Y | N | SRe2L  | 15.6 | 22.9 | 54.8 |
> |      | Y  | Y  | +ours  | **21.4** | **33.2** | **58.4** |
>
> These results indicate that even when both the distilled images and soft labels are biased (i.e., derived from long-tailed models), our method still outperforms a baseline that uses soft labels from unbiased distillation model. This highlights the effectiveness of our method in mitigating the impact of image-induced bias.
>
> We do not follow the baseline setting (biased images with unbiased labels) because our method requires class-frequency information, determined by the imbalance factor (IF), from the distillation model's training distribution. Therefore, a model trained on balanced data is incompatible. This design also reflects realistic scenarios, where both the distillation model and synthetic images originate from the same long-tailed dataset.
>
> We will revise the main text to better highlight this point and include this experiment in Sec. 4.2 of the revised version.
>
> **Q2: In Figure 3(b), why is there a sudden performance improvement at the end, even as the imbalance factor increases?**
>
> **A:** The direct cause lies in our use of SRe2L’s default hyperparameters - specifically, the number of training epochs (EP=200) for the distillation model, which resulted in suboptimal performance. To investigate this, we compared the performance of synthetic datasets distilled using **models trained with different numbers of epochs** on CIFAR10:
>
> Exp1: Effect of Model Epoch on Performance with IF=10
>
> | Epoch | IF   | Method | IPC=10    | IPC=50    |
> | - | - | - | - | - |
> | 10    | 10   | SRe2L  | 39.6 | 55.0 |
> |       |      | +ours  | 41.9  | 63.2 |
> | 30    | 10   | SRe2L  | 27.2    | 51.5    |
> |       |      | +ours  | 26.2    | 50.0  |
> | 50    | 10   | SRe2L  | 24.6      | 46.8     |
> |       |      | +ours  | 25.2     | 53.6    |
>
> Exp2: Effect of Model Epoch on Performance with IF=100
>
> | Epoch | IF   | Method | IPC=10    | IPC=50    |
> | - | - | - | - | - |
> | 10    | 100  | SRe2L  | 21.6    | 23.6     |
> |       |      | +ours  | 33.0 | 37.9    |
> | 30    | 100  | SRe2L  | 28.1  | 38.2    |
> |       |      | +ours  | 32.0     | 59.1 |
> | 50    | 100  | SRe2L  | 23.9    | 40.3 |
> |       |      | +ours  | 29.1     | 53.8    |
>
> We observe that as the epoch increases, the performance of the distilled dataset first improves and then declines, indicating the existence of an optimal epoch. Moreover, this optimal epoch tends to increase as the imbalance factor (IF) or images per class (IPC) increases. This means that performance of distilled dataset does not monotonically align with accuracy of distillation model. As shown in [1], early-stage models provide richer soft labels, while late-stage models yield overconfident outputs. Similarly, [2–3] show that proper model epoch selection is crucial for effective trajectory matching.
>
> In the original setting, we use fixed training epochs (200) for the distillation model. As the imbalance factor (IF) increases, the optimal epoch shifts later, which explains the seemingly anomalous improvement in performance with higher imbalance.
>
> We revised our experimental setup by selecting the best-performing model from $\text{Epoch}\in\set{10,30,50}$, and replotted Figure 3(b) accordingly. As shown, the previously observed anomalous performance trend disappears.
> The updated results are presented below.
>
> | tau\IF |        10 |        50 |       100 |       500 |      1000 |      2000 |
> | - | - | - | - | - | - | - |
> |      0 |     32.6 |     28.2 |    29.2 |     23.9 |     21.9 |    14.5 |
> |      1 |    39.7 |     33.5 |     34.0 |     30.1 |     25.3 |     18.9 |
> |    1.5 |     40.6 |    33.3 |     34.4 |    28.8 |    28.2 |    20.5 |
> | ours | **41.6** | **42.7** | **35.8** | **32.5** | **29.3** | **27.4** |
>
> We will provide all experimental results in the appendix and update the corresponding results in the paper. Additional results are available upon request during the discussion, given the rebuttal’s word limit.
>
> **Q3: ADSA performance sometimes  drops with small IPC. Why?**
>
> **A:** As noted in Section 3.1, low IPC makes it difficult to capture the original distribution, weakening alignment between labels and images. In addition, our method depends on stable soft-label statistics for calibration, which become unreliable with very limited data.
>
> **Q4: In Fig. 1(b), the balanced model seems to generate images, not just relabel.**
>
> **A:** Thank you for pointing out the error in the figure. The labels below the models in Fig. 1(b) were mistakenly swapped and will be corrected in the revised version.
>
> **Q5: When applied to a balanced training dataset, does ADSA lead to any performance degradation?**
>
> **A:** ADSA does not cause any performance degradation when applied to a balanced training dataset, which follows directly from its formulation. Specifically, the calibration mechanism defined in Eq. (9) is:
> $$
> p(y|x; \tau) = \frac{\exp(f_y(x) - \tau \log \pi_y)}{\sum_{y' \in [C]} \exp(f_{y'}(x) - \tau \log \pi_{y'})},
> $$
> where $f_y(x)$ denotes the predicted logit for class $y$, and $\pi_y$ represents the class prior (i.e., the frequency of class $y$ in the training data).
>
> In the case of a balanced dataset with $K$ classes, we have $\pi_1 = \pi_2 = \cdots = \pi_K = \frac{1}{K}$. Substituting into the equation yields:
> $$
> p(y|x; \tau) = \frac{\exp(f_y(x) + \tau \log K)}{\sum_{y' \in [C]} \exp(f_{y'}(x) + \tau \log K)}
> = \frac{\exp(f_y(x))}{\sum_{y' \in [C]} \exp(f_{y'}(x))}
> $$
>
> The calibration term cancels out as a constant, so our method reduces to standard distillation and causes no change in balanced settings.
>
> **Q6: Does ADSA introduce computational or memory bottlenecks as data and class counts grow?**
>
> **A:** We fully understand your concern. However, the computational and memory overhead introduced by our method is minimal and negligible compared to baseline methods, and it imposes no practical burden in real-world applications. We simulate runtime under varying numbers of classes and IPCs using ResNet-18, with time measured in **seconds**.
>
> | Resolution | CLASS_NUM \ IPC | IPC=1 | IPC=10 | IPC=50 | IPC=100 |
> | - | - | - | - | - | - |
> | 32         | 10              | 1.7  | 1.7  | 2.0     | 2.0 |
> | 32         | 100             | 1.7  | 2.2  | 3.3   | 4.7   |
> | 224        | 1000            | 7.8  | 29.5 | 107.2 | 205.3 |
>
> We only store logits of shape $\text{IPC} \times \text{CLASS} \times \text{CLASS}$, requiring minimal GPU memory. Compared to standard dataset distillation baselines, our overhead remains insignificant, as the table shown below.
>
> | IPC  | CLASS_NUM | FP32 (4 Bytes) |
> | - | - | - |
> | 1    | 10    | 400 Bytes      |
> | 1    | 100   | 39.1 KB      |
> | 1    | 1000  | 3.8 MB        |
> | 10   | 10    | 3.9 KB        |
> | 10   | 100   | 390.6 KB      |
> | 10   | 1000  | 38.2 MB      |
> | 50   | 10    | 19.5 KB       |
> | 50   | 100   | 1.9 MB        |
> | 50   | 1000  | 190.7 MB      |
>
> Refs:
>
> [1] A Label Is Worth a Thousand Images in Dataset Distillation, NIPS, 2024
>
> [2] Towards Lossless Dataset Distillation via Difficulty-Aligned Trajectory Matching, ICLR, 2024
>
> [3] Dataset Distillation by Automatic Training Trajectories, ECCV, 2024

---

> > ### Comment · Reviewer_cicZ · 2025-08-05
> >
> > After carefully revisiting both the authors’ rebuttal and the main paper, I acknowledge that the authors have addressed most of my concerns—particularly those related to bias in distilled images and certain empirical inconsistencies in the main paper. I also appreciate the clarification regarding the applicability to balanced datasets and the potential computational or memory bottlenecks as the number of data points or classes increases.
> >
> > While I am not an expert in dataset distillation, I believe the authors have provided the best possible clarification within the constraints of the rebuttal phase, and their responses appear constructive.

---

> > > ### Author Response · Authors · 2025-08-05
> > >
> > > Thank you so much for your encouraging and constructive feedback! We'll continue our efforts to further advance the dataset distillation community.

---

### Official Review · Reviewer_Prxx · 2025-06-29

**Clarity:** 2
**Significance:** 3
**Originality:** 3
**Rating:** 5
**Confidence:** 3

**Summary:**

This paper proposes a lightweight module, ADSA, to calibrate soft-label biases in long-tailed dataset distillation, which stem from both the teacher model and the synthetic images. By deriving an imbalance-aware generalization bound, the authors reveal the underlying causes of performance degradation. Extensive experiments demonstrate that ADSA consistently improves performance across various distillation methods and label budgets, achieving up to an 11.8% improvement in tail-class accuracy on ImageNet-1k-LT.

**Questions:**

1.Equation (3): The constant C is undefined. Could the authors clarify its meaning and boundary conditions in this context?

2.Equation (15): It assumes ptr​ (x∣y)=pte(x∣y), a common setting in long-tailed learning. Does this assumption still hold in dataset distillation scenarios?

3. Are the reported results from a single run? The absence of variance or confidence intervals makes it difficult to assess stability. It is recommended to report results over multiple random seeds.

If the authors are able to address the above issues, I would be willing to raise my score.

**Ethical Concerns:**

["NO or VERY MINOR ethics concerns only"]

**Final Justification:**

The author has addressed most of my questions, including the setting of variance, etc.

**Limitations:**

yes

**Quality:**

3

**Strengths And Weaknesses:**

Strengths

The proposed lightweight module effectively mitigates these biases and delivers consistent performance improvements across various settings. Experimental results show that the method outperforms existing strong baselines.

Weaknesses

1.In Equations (1)–(2), the distilled dataset is denoted as 𝐷syn​ , but in Equations (4)–(6), it suddenly switches to 𝐷dd  (e.g., Rdd). The notation should be unified or clearly explained.

2.LAD is presented as a comparative long-tailed distillation method, but its results are missing under some IPC/IF settings (e.g., IPC=1 is absent in Table 1). If feasible, these should be included for completeness.

3.The experiments only report the hardware used, but lack details such as per-experiment runtime and memory usage (e.g., GPU hours required to distill ImageNet-LT with IPC=50), which limits reproducibility and practical cost assessment.

---

> ### Author Rebuttal · Authors · 2025-07-31
>
> We sincerely thank the reviewer for the careful and detailed comments on our paper, including the experimental design and completeness. In response, we have conducted extensive additional experiments and provided detailed clarifications and revisions as outlined below.
>
> **W1: The notation about $D_{dd}$ or $D_{syn}$ should be unified or clearly explained.**
>
> **A:** Thank you for pointing out the issue with the notation. We will revise the manuscript to unify the notation by consistently using the subscript $dd$, such as $D_{dd}$ and $\theta_{dd}$, throughout the paper for clarity and consistency.
>
> **W2: Completion of LAD[1].**
>
> **A:** We have reproduced the results of LAD and reported the corresponding variances, as summarized in the table below:
>
> Reproduction of LAD under Varying IPC and Imbalance Factors on CIFAR-10
>
> | IPC\IF | 10       | 50       | 100      |
> | ------ | -------- | -------- | -------- |
> | 1      | 28.0±1.0 | 24.4±0.3 | 23.8±0.5 |
> | 10     | 58.1±0.3 | 54.2±1.0 | 53.4±0.1 |
> | 50     | 70.5±0.4 | 65.8±0.2 | 64.0±0.9 |
>
> Reproduction of LAD under Varying IPC and Imbalance Factors on CIFAR-100
>
> | IPC\IF | 10       | 50       | 100      |
> | ------ | -------- | -------- | -------- |
> | 1      | 12.3±0.2 | 11.1±0.1 | 10.6±0.1 |
> | 10     | 31.5±0.2 | 26.8±0.3 | 24.9±0.1 |
> | 50     | 40.0±0.1 | 34.5±0.1 | 31.6±0.0 |
>
> **W3: Details about per-experiment runtime and memory usage.**
>
> **A:**  We conducted our experiments on a server equipped with 8×RTX 4090 GPUs. Due to the relatively small scale of CIFAR10 and CIFAR100, the GPU utilization may not be fully saturated. As a result, their memory usage is approximately similar, and CIFAR10 runs slightly faster than CIFAR100. The approximate computational cost and GPU memory footprint for both CIFAR10/CIFAR100 are summarized below:
>
> |            | SRe2L | GVBSM | EDC  |
> | --------------- | ----- | ----- | ---- |
> | GPU Memory (GB) | 2     | 2     | 2    |
> | Time (GPU·h)    | 2     | 10    | 5    |
>
> The conservative runtime estimates on ImageNet are as follows:
>
> |                 | SRe2L     | GVBSM      | EDC        |
> | --------------- | --------- | ---------- | ---------- |
> | GPU Memory (GB) | 10        | 20         | 20         |
> | Time (GPU·h)    | <40 (+16) | <360 (+80) | <180 (+80) |
>
> The values in parentheses represent the extra GPU hours needed to re-train the distillation models on long-tailed datasets, since the public checkpoints from Torch are pre-trained on balanced ImageNet and thus unsuitable for a fair evaluation. We adopt the default batch sizes used in SRe2L, EDC, and GVBSM. The backbone architectures have been detailed in Table 5 of Appendix A.2. Moreover, we observe that during the soft-label generation phase, the GPU utilization of SRe2L, GVBSM, and EDC tends to be relatively low, indicating potential for further acceleration.
>
> **Q1: Clarify the meaning and boundary conditions about the constant $C$.**
>
> **A:** The constant $C$ represents an upper bound on the negative log-likelihood of the model $\hat{p}(y|x)$, which is typically finite in practical training scenarios.
>
> Let $\hat{p}(y|x)$ denote the predictive distribution of any classifier, typically referring to the output of a trained network on the distribution $p_{dd}(x, y)$. If the negative log-likelihood $-\log \hat{p}(y|x)$ is upper-bounded by a positive constant $C$, then we have the following inequality (as shown in Eq. 3) according to D3S[2]:
> $$
> l_{tr} \le l_{dd} + \frac{C}{2\sqrt{2}} \sqrt{R_{dd}}, \\\\
> R_{dd} = D_{KL}(p_{tr}(x)\|p_{dd}(x)) + D_{KL}(p_{tr}(y | x)\|p_{dd}(y | x)).
> $$
>
> Here, $l_{tr} = E_{p_{tr}}[-\log \hat{p}(y|x)]$ and $l_{dd} = E_{p_{dd}}[-\log \hat{p}(y|x)]$. The distributions $p_{tr}$ and $p_{dd}$ correspond to the data sampling distributions from the original training set $D_{tr}$ and the distilled dataset $D_{dd}$, respectively.
>
> **Q2: Equation (15): It assumes $p_{tr}(x|y) = p_{te}(x|y)$, a common setting in long-tailed learning. Does this assumption still hold in dataset distillation scenarios?**
>
> **A:** Yes, this assumption still holds. In long-tailed learning, the training set is a long-tailed dataset $D_{tr}$, and the test set is a balanced dataset $D_{te}$, satisfying $p_{tr}(x|y) = p_{te}(x|y)$.
>
> In long-tailed dataset distillation, the original training set is also the long-tailed dataset $D_{tr}$, from which a distilled dataset $D_{dd}$ is derived. The model is trained on $D_{dd}$ and evaluated on the same balanced test set $D_{te}$.
>
> Since the original training and test sets used in our distillation setting are identical to those in long-tailed learning, the assumption $p_{tr}(x|y) = p_{te}(x|y)$ remains valid in the dataset distillation scenario.
>
> **Q3: The absence of variance or confidence intervals makes it difficult to assess stability.**
>
> **A:** We conducted three runs of experiments on CIFAR10 and CIFAR100, and report the mean and standard deviation in the tables below.
>
> Experiment results under Varying IPC and Imbalance Factors on CIFAR-10
>
> | IPC        | 1            | 1            | 1            | 10           | 10           | 10           | 50           | 50           | 50           |
> | ---------- | ------------ | ------------ | ------------ | ------------ | ------------ | ------------ | ------------ | ------------ | ------------ |
> | methods\IF | 10           | 50           | 100          | 10           | 50           | 100          | 10           | 50           | 100          |
> | LAD        | 28.0±1.0     | 24.4±0.3     | 23.8±0.5     | 58.1±0.3     | 54.2±1.0     | 53.4±0.1     | 70.5±0.4     | 65.8±0.2     | 64.0±0.9     |
> | SRe2L      | 18.0±0.5     | 20.1±0.6     | 17.2±0.6     | 39.1±0.7     | 27.0±0.5     | 28.1±0.1     | 55.1±0.2     | 51.4±1.0     | 40.1±0.4     |
> | SRe2L+ours | **22.1±1.6** | **26.6±1.0** | **23.8±0.6** | **41.5±0.6** | **35.5±0.4** | **33.0±0.1** | **63.3±0.1** | **63.8±0.7** | **58.8±0.5** |
> | GVBSM      | 34.2±0.4     | 28.9±0.4     | 25.1±0.3     | 49.5±0.1     | 33.8±0.3     | 29.4±0.1     | 58.2±0.2     | 37.2±0.0     | 30.9±0.0     |
> | GVBSM+ours | **39.1±0.4** | **36.0±0.6** | **31.5±0.5** | **54.4±0.5** | **45.8±0.1** | **40.4±0.4** | **64.7±0.0** | **51.4±0.2** | **46.9±0.2** |
> | EDC        | 32.3±1.4     | 29.5±0.8     | 30.0±0.6     | 69.9±0.7     | 58.5±0.5     | 50.9±0.5     | 77.8±0.1     | 65.6±0.3     | 56.0±0.2     |
> | EDC+ours   | **35.2±0.8** | **36.4±1.0** | **39.3±0.6** | **73.2±0.4** | **69.8±0.2** | **68.7±0.3** | **80.8±0.2** | **76.4±0.4** | **74.8±0.1** |
>
> Experiment results under Varying IPC and Imbalance Factors on CIFAR-100
>
> | IPC        | 1            | 1            | 1            | 10           | 10           | 10           | 50           | 50           | 50           |
> | ---------- | ------------ | ------------ | ------------ | ------------ | ------------ | ------------ | ------------ | ------------ | ------------ |
> | methods\IF | 10           | 50           | 100          | 10           | 50           | 100          | 10           | 50           | 100          |
> | LAD        | 12.3±0.2     | 11.1±0.1     | 10.6±0.1     | 31.5±0.2     | 26.8±0.3     | 24.9±0.1     | 40.0±0.1     | 34.5±0.1     | 31.6±0.0     |
> | SRe2L      | 17.3±0.9     | 17.5±0.2     | 15.7±0.3     | 35.4±0.2     | 30.1±0.1     | 15.7±0.3     | 46.1±1.1     | 35.0±0.2     | 31.8±0.7     |
> | SRe2L+ours | 17.2±0.3     | **18.1±0.1** | 9.3±0.2      | **38.2±0.0** | **34.2±0.1** | **33.0±0.0** | **53.7±0.1** | **45.3±0.2** | **41.7±0.1** |
> | GVBSM      | 20.3±0.3     | 19.7±0.1     | 19.1±0.2     | 35.1±0.0     | 31.2±0.2     | 27.9±0.1     | 39.2±0.3     | 34.0±0.1     | 31.1±0.1     |
> | GVBSM+ours | 20.0±0.3     | **20.4±0.3** | 17.0±0.6     | **35.5±0.1** | **32.8±0.1** | **29.6±0.3** | **40.1±0.0** | **35.8±0.3** | **33.2±0.4** |
> | EDC        | 42.3±0.0     | 34.3±0.3     | 32.0±0.1     | 54.3±0.6     | 43.4±1.6     | 39.0±1.8     | 57.0±0.8     | 45.7±1.6     | 40.9±1.8     |
> | EDC+ours   | **43.4±0.4** | **37.9±0.5** | **34.1±0.3** | **55.5±0.1** | **46.0±0.1** | **41.6±0.1** | **58.0±0.2** | **48.1±0.1** | **43.5±0.2** |
>
> We corrected the number of training epochs used for the distillation model in SRe2L, which leads to slightly better performance compared to the original paper. During the rebuttal process, we found that using an early-stage distillation model can significantly improve the quality of the distilled dataset. We will update the revised version with the corrected results accordingly. Nevertheless, the original results remain meaningful, as our method is an add-on approach that focuses on relative improvements.
>
> Refs:
>
> [1] Distilling long-tailed datasets, CVPR, 2025
>
> [2] Large Scale Dataset Distillation with Domain Shift, ICML, 2024

---

> > ### Comment · Reviewer_Prxx · 2025-08-05
> >
> > Thank you for your detailed reply, author. I believe this is crucial for measuring the workload, so I've given a positive score.

---

> ### Author Response · Authors · 2025-08-05
>
> Thank you again for your valuable suggestions regarding our baselines and experiments, and for your support of our work. We will further refine the current version of the paper.

---

### Official Review · Reviewer_GTsB · 2025-07-01

**Clarity:** 2
**Significance:** 3
**Originality:** 3
**Rating:** 4
**Confidence:** 3

**Summary:**

In this paper, authors study dataset distillation problem in the setting where the dataset follows a long-tailed distribution. First, they conduct a theoretical and empirical analysis, showing that the bias comes from both distilled images as well the distillation model. Based on this observation, the authors then propose adaptive soft label alignment module. In this proposed method, they adopt a logit calibration method. They show that the logit calibration method can be applied to existing dataset distillation methods in a post-hoc manner. Their experiment results show that their method effectively mitigate the poor performance issued caused by long tailed datasets.

**Questions:**

1. I am a bit confused about the experiment setup. My current understanding is that they authors frame their method as a simple add on of existing dataset distillation pipeline. And in their main experiments (Section 4.1 Table 1, 2), they show existing methods performance (such as SRe2L, EDC) and their add on. Just to clarify, authors trained all the experts, distilled all the images from the long-tailed distribution, the "ours" part is to apply Eq. 9 and 10 to soft labels. In other words, during the entire dataset distillation pipeline, there is no access to balanced datasets.


2. I am also quite confused about Figure 1(a), why does original dataset equal imbalanced dataset plus balanced dataset? Overall, I think this paper could benefit from more clear writing about the set up.

**Ethical Concerns:**

["NO or VERY MINOR ethics concerns only"]

**Final Justification:**

During rebuttal, authors have addressed my concerns regarding baselines. They also clarified a mis-understanding I have regarding the methodology and after the clarification, I think their method makes sense.

**Limitations:**

Yes, authors have addressed the limitations and potential negative societal impact.

**Quality:**

2

**Strengths And Weaknesses:**

**Strengths**

   1. Well-motivated problem:

The paper tackles a practically important issue—real-world datasets often exhibit class imbalances rather than a clean uniform distribution. Addressing dataset distillation specifically in long-tailed settings is clearly relevant and should interest the broader research community.

   2. Insightful empirical analysis:

The paper provides a careful and insightful empirical analysis, clearly illustrating when and why existing dataset distillation methods fail under long-tailed distributions. The use of three distinct configurations to identify and isolate the sources of bias is particularly effective and enhances understanding of the underlying problem.

**Weaknesses**

   1. Incomplete handling of identified biases:

My understanding is that the central methodological contribution is the introduction of Eq. 9, which essentially applies a frequency-based penalization to mitigate bias in soft labels. While this approach logically addresses bias introduced by class imbalance in the training set, the authors neglect the bias arising specifically from the distilled images, even though it was explicitly highlighted as a key issue in the empirical analysis (Section 3.2). The paper would benefit from either a justification of why the image-induced bias is less critical or a discussion of possible solutions for addressing this image-specific bias.

   2. Lack of essential baseline:

Training directly on class-imbalanced datasets is known to lead to suboptimal performance even beyond the dataset distillation setting. An intuitive baseline would be to apply well-established data balancing techniques (such as oversampling or weighted sampling) to the original long-tailed dataset prior to distillation. Such a baseline would clarify whether simple and well-known rebalancing methods could already sufficiently mitigate the issue without requiring specialized adjustments in dataset distillation methods.

---

> ### Author Rebuttal · Authors · 2025-07-31
>
> We sincerely thank the reviewer for the insightful suggestions on both the content and experimental aspects of our paper. In response, we have conducted additional experiments and carefully addressed each point in the replies below.
>
> **W1: Incomplete handling of identified biases. Central methodological contribution only mitigate the bias introduced by labels from distillation model, neglecting the bias from the distilled images.**
>
> **A:** Our proposed method could mitigate the two bias sources (distilled images and distillation model) in soft label. A key contribution of our method is its ability to implicitly rectify both sources of bias on soft labels including distilled image bias and soft-label bias in a unified manner. We conducted two additional experiments to prove that our method could mitigate both the bias succesfully.
>
> Exp1: Addressing bias from the distillation model (biased soft labels)
>
> We use soft labels produced by a model trained on an long-tailed dataset, while distilled images are obtained from a model trained on a balanced dataset.
>
> | IF   | biased img | biased label | method | IPC=1    | IPC=10   | IPC=50   |
> | ---- | ---------- | ------------ | ------ | -------- | -------- | -------- |
> | 10   | N          | Y            | SRe2L  | 16.6     | 34.3     | 50.6     |
> |      | N          | Y            | +ours  | **19.0** | **41.4** | **62.6** |
> | 100  | N          | Y            | SRe2L  | 18.3     | 24.4     | 31.5     |
> |      | N          | Y            | +ours  | **21.8** | **36.9** | **57.6** |
>
> > Note: "Y" indicates the presence of bias; "N" indicates no bias in that source. "IF" means Imbalance Factor.
>
> These results clearly show that our method can effectively mitigate soft-label bias introduced by a biased distillation model.
>
> Exp2: Addressing bias in distilled images
>
> This experiment evaluates the bias introduced by distilled images. The baseline method SRe2L uses images from a long-tailed model and soft labels from a balanced one, while our method is applied when both images and labels are from long-tailed models.
>
> | IF   | biased img | biased label | method | IPC=1    | IPC=10   | IPC=50   |
> | ---- | ---------- | ------------ | ------ | -------- | -------- | -------- |
> | 10   | Y          | N            | SRe2L  | 16.0     | 34.7     | 55.3     |
> |      | Y          | Y            | +ours  | **23.7** | **40.2** | **63.1** |
> | 100  | Y          | N            | SRe2L  | 15.6     | 22.9     | 54.8     |
> |      | Y          | Y            | +ours  | **21.4** | **33.2** | **58.4** |
>
> These results indicate that even when both the distilled images and soft labels are biased (i.e., derived from long-tailed models), our method still outperforms a baseline that uses soft labels from unbiased distillation model. This highlights the effectiveness of our method in mitigating the impact of image-induced bias.
>
> Ours method do not follow the baseline setting (biased images with unbiased labels) because our method requires class-frequency information, determined by the imbalance factor (IF), from the distillation model's training distribution. Therefore, it is incompatible to apply our method on a model trained with balanced dataset. This design also reflects realistic scenarios, where both the distillation model and synthetic images originate from the same long-tailed model.
>
> We will revise the main text to better highlight this point and include this experiment in Sec. 4.2 of the revised version.
>
> **W2: Lack of essential baseline. An intuitive baseline would be to apply well-established data balancing techniques (such as oversampling or weighted sampling) to the original long-tailed dataset prior to distillation.**
>
> **A**:  We have conducted the additional experiments about enhance baseline and will include the results in the revised version of the paper, along with releasing the corresponding baseline code.
>
> Experiment Setup: We applied a resampling technique (random oversampling) to the long-tailed dataset prior to training the distillation model. The rest of the distillation pipeline remains unchanged. Below are the results for CIFAR-10 and CIFAR-100 under various imbalance factors (IF), image per class (IPC), and whether our method is used.
>
> Exp1: Experiment on CIFAR-10, we use SRe2L as our baseline method.
>
> | IF      | method          | IPC=1    | IPC=10   | IPC=50   |
> | ------- | :--------------- | -------- | -------- | -------- |
> | **10**  | SRe2L           | 19.9     | 39.0     | 55.2     |
> |         | +resample | 18.1     | 38.1     | 54.8     |
> |         | +ours         | **23.7** | **40.2** | **63.1** |
> |         | +resample+ours | 19.5     | 38.0     | 58.8     |
> | **50**  | SRe2L           | 19.9     | 27.2     | 51.1     |
> |  | +resample     | 21.7     | 41.3     | 54.6     |
> |         | +ours | **26.9** | 35.4     | **63.7** |
> |         | +resample+ours | 22.3     | **44.1** | 56.6     |
> | **100** | SRe2L   | 18.7     | 21.3     | 24.0     |
> |         | +resample     | **25.5** | 36.4     | 49.5     |
> |  | +ours | 21.4     | 33.2     | **58.4** |
> |         | +resample+ours | 24.1     | **37.1** | 56.1     |
>
> Exp2: Experiment on CIFAR-100, we use SRe2L as our baseline method.
>
> | IF   | method        | IPC=1    | IPC=10   | IPC=50   |
> | ---- | :------------- | -------- | -------- | -------- |
> | 10   | SRe2L  | 16.3     | 35.8     | 45.2     |
> |      | +resample | **20.3** | 35.8     | 45.1     |
> |      | +ours | 16.9     | **38.2** | **53.9** |
> |      | +resample+ours | 18.5     | 37.6     | 50.6     |
> | 50   | SRe2L  | 17.8     | 28.8     | 33.5     |
> |      | +resample     | **18.9** | 29.2     | 36.1     |
> |      | +ours | 18.2     | **34.2** | **45.6** |
> |      | +resample+ours | 17.6     | 32.2     | 39.6     |
> | 100  | SRe2L  | 15.9     | 29.1     | 31.8     |
> |      | +resample     | **16.3** | 28.4     | 32.8     |
> |      | +ours | 9.6      | **33.0** | **41.7** |
> |      | +resample+ours | 15.1     | 29.9     | 34.3     |
>
> >  Note: The original paper used SRe2L with its default distillation model training epoch (EP = 200). During the rebuttal process, we found this choice does not always yield optimal performance. Therefore, for fairness, we re-evaluated SRe2L with resampling or our methods, using the best-performing distillation model, selected from $\text{EP}\in\set{10,30,50}$ for CIFAR-10, and $\text{EP}\in\set{10,50,100,200}$ for CIFAR-100. We will update the reported SRe2L results in the main paper and provide the full version in the appendix. If needed, we are happy to share the full results during the discussion phase.
> >
>
> The experimental results show that our method outperforms both the baseline and the resample setting in most cases, while resampling indeed improves over the baseline. The results show that our method could be seamlessly integrated as a plug-and-play module to further boost the performance of resampling-based pipelines. This highlights the generality and robustness of our approach across various settings.
>
> The relatively lower performance under IPC = 1 may be attributed to the limited number of samples, which makes it difficult to reliably estimate the class-level soft-label imbalance. This in turn affects the accuracy of our calibration.
>
> **Q1: Confusion about the experiment setup. The "ours" part is to apply Eq. 9 and 10 to soft labels. In other words, during the entire dataset distillation pipeline, there is no access to balanced datasets.**
>
> **A**: Your understanding is correct. Yes, there is no access to any balanced dataset during the entire distillation pipeline. All baselines, including our add-on method, are trained only on the long-tailed dataset and evaluated on a balanced test set.
>
> In Sec. 3.2, we use a balanced dataset only for analysis purposes, to isolate and identify the two sources of soft-label bias. However, in Sec. 3.3 and all experimental evaluations, the method is trained strictly on long-tailed data, aligning with realistic settings.
>
> **Q2: Clarification on Figure 1(a), why does original dataset equal imbalanced dataset plus balanced dataset?**
>
> **A:** Thank you for your thoughtful observation. The original dataset is *not* the sum of the imbalanced and balanced datasets. In Figure 1(a), our intention is to illustrate that the original dataset in dataset distillation can be either *imbalanced* or *balanced*. While many prior works in dataset distillation assume a balanced original dataset, real-world data often follows a long-tailed (imbalanced) distribution.

---

> ### Author Response · Authors · 2025-08-05
> **Kindly Request Feedback from Reviewer GTsB**
>
> Dear Reviewer GTsB,
>
> We sincerely appreciate the time and effort you devoted to reviewing our manuscript. In response to your thoughtful feedback, we have submitted a rebuttal with extensive experimental results addressing your concerns. The key updates include:
>
> * **Effectiveness of ADSA in addressing both sources of bias:** We conducted experiments showing that ADSA effectively mitigates bias arising from both the distillation model and the distilled images.
> * **Comparison with a resampling-based baseline:** Following your suggestion, we added a baseline using a resampling strategy. Experimental results show that our method outperforms this baseline, and can also serve as a plug-and-play module to further enhance its performance.
> * **Clarification on experimental setup:** Our method does not access any balanced dataset throughout the entire long-tailed dataset distillation pipeline.
> * **Clarification on Figure 1(a):** The original dataset used in this work is long-tailed. Figure 1(a) is intended to illustrate that, in real-world scenarios, data distributions are typically long-tailed rather than ideally uniform. We have revised the description accordingly in the updated version to improve clarity.
>
> We hope that these clarifications and additional experiments help address your concerns. We welcome any further feedback or questions. Thank you again for your invaluable feedback and for considering our response during the rebuttal process.
>
> Sincerely,
>
> The Authors

---

> > ### Comment · Reviewer_GTsB · 2025-08-05
> >
> > Thank you so much for your detailed response! You have addressed my concerns regarding baselines and my understanding on the set up. I will increase my score accordingly!

---

> ### Author Response · Authors · 2025-08-05
>
> We sincerely appreciate your valuable suggestions on our paper. We will revise the descriptions and include additional experiments in the final version. Thank you for helping us improve the quality of our work. We will continue striving to contribute more to the dataset distillation community.

---

### Official Review · Reviewer_bfAk · 2025-07-02

**Clarity:** 2
**Significance:** 3
**Originality:** 2
**Rating:** 4
**Confidence:** 2

**Summary:**

The paper addresses the problem of distilling long-tailed dataset information. In particular, the authors derive a generalization error bound by adapting the formulation used in D3S [1] for the long-tailed setting under label shift. Based on that bound, it is argued that the distilled data can introduce bias through two sources: (i) the distilled images and (ii) their soft-labels, while their effect is evaluated via a perturbation analysis controlling different conditions. To account for these biases, post-hoc logit-adjustment (LA) [2] is used to correct soft-labels of the distilled dataset. Having observed a distribution shift between real and synthetic data distributions, the authors propose to minimize class-wise confidence variance to calibrate the strength factor of LA, leveraging distilled data. This adaptive soft label alignment leads to improvement in the overall and tail classification performance across different imbalance settings and SOTA data-distillation methods.

Extra references:

[1] Loo, Noel, et al. "Large scale dataset distillation with domain shift." Forty-first International Conference on Machine Learning. 2024.

[2] Menon, Aditya Krishna, et al. "Long-tail learning via logit adjustment." arXiv preprint arXiv:2007.07314 (2020).

**Questions:**

- Q1: Can you include the intermediate steps when deriving the forms 1 and 2 when proving Th 3.1. to ensure correctness?

- Q2: I can follow point (iii) in L175 in relation to the second term in Eq 5.but I do not follow why this has to hold for each class when looking at the second term of Eq 6. e.g., miss-labelled but identical samples could still minimize the second term in Eq 6.

- Q3: I found the connection between the analysis in Sec 3.2. and the insights drawn from Th 3.1. to be unclear.
   - Q3.1: Could you provide a connection between forms 1 and 2 with the three configs considered? e.g., clarify which term each configuration is trying to optimize. Also, including a config (4) with balanced/balanced would make for a more complete perturbation analysis.
   - Q3.2: The generalization bound is derived on the full test loss, in that sense it would be interesting to see what happens to the overall performance (overall tail+head performance)
   - Q3.3:  L242-244. This finding contradicts the theory developed which calls for a deeper analysis on this matter.
   - Q3.4:  L212-214. Shouldn’t we draw that conclusion when comparing config 1 with config 3 instead where they only differ by the DM (2)?
   - Q3.5: When solving Eq. 7. across the three configs, are you using the same g network and only ablate the f_{tr}?

Q4: Fig3. (b) The proposed method does not display a decreasing performance trend as the IF increases in contrast to the baselines. Could you explain why this is the case?

Typos/Minor:
(I am not expecting an answer, please reflect upon these and refine if necessary)

- T1. Define what the ADSA abbreviation means the first time you use it in the main text.
- T2. L146. is \theta_{syn}== \theta_{dd}?
- T3. Eq.3 the C is not defined.
- T4. Eq.5 define the connection between the densities $p$ and the datasets. E.g. how is the p_tr related to D_tr?
- T5. L150. did you mean training loss?
- T6. Definition of Th 3.1. lacks rigor, what is C? Under what assumption Th 3.1. holds? The Th 3.1. as provided in [1] is a good reference point.
- T7. Tab 1. SRe2L + ours (IP=1 and IF=50) should not be in bold.
- T8. Fig 1. (b) please define the DM (1) and DM (2) abbreviations. Also it appears that (only for relabelling and generating images have been mixed).
- T9. Fig3. Left panel, are these the correct titles? Right panel, what is the IF used for (a) and the IPC for (b)?
- T10. L573.The first paragraph and second paragraph overlap significantly.
- T11. L596. Please provide some more details on how these t-SNE plots were constructed. e.g., are you using the same feature extractor for all settings?
- T12. L317. define what the soft-label budget means to make the paper self-contained.
- T13. Eq.10 you could use a different index (e.g., j) on the inner summation to avoid confusion.

Final comments:
Although the empirical results are convincing, the proposed method appears to be a trivial extension of logit-adjustment in the dataset condensation and is disconnected from the theory developed, limiting the depth of the study. Although the developed theory is interesting, the empirical results do not always support it, calling for further investigation. Concluding, I do not see how the theoretical insights connect with the proposed method, which nevertheless appears to perform well by using existing methodology (i.e., logit-adjustment).

**Ethical Concerns:**

["NO or VERY MINOR ethics concerns only"]

**Final Justification:**

Despite my initial concerns, the authors have made a substantial effort in their rebuttal, providing complementary theoretical derivations and additional experiments. While I still have some minor doubts regarding the connections between theory-perturbation analysis-method, I am now persuaded of the following:

* The theory is sound in both of its alternative formulations.
* Although the perturbation analysis remains somewhat tangential to the main theory, it effectively disentangles bias sources in practical terms (i.e., distinguishing how the long-tailed bias affects the generative and discriminative components).
* The method—while not entirely novel—flexibly extends the original logit-adjusted approach and, for the first time, applies it to dataset distillation; it performs well under both types of bias, as demonstrated by the new experiments.
* It can be used as an add-on to previous baselines that focus primarily on feature alignment, with consistent improvements.
* The authors have established a partial theoretical connection between the method and the underlying theory, and have empirically demonstrated that refining the pseudo-labels corrects both the marginal and posterior terms.

On balance, these strengths outweigh my remaining reservations, and I am therefore raising my score.

**Limitations:**

yes

**Paper Formatting Concerns:**

No issues detected

**Quality:**

2

**Strengths And Weaknesses:**

Strengths:
- Partial theoretical connection between the generalization bound and imbalance levels.
- The experimental setting convincingly demonstrates that the proposed adaptive logit-adjustment applies to long-tailed dataset-distillation and boosts the overall performance across different baselines and imbalance settings.
- The proposed method is simple and can be easily applied on top of methods originally developed for balanced dataset-distillation.

Weaknesses:
- W1 - The connection between the forms in Eqs. 5 and 6 and the perturbation analysis is unclear.
- W2 - The developed theory attributes tail performance collapse to two different sources, yet the proposed method is disconnected from this finding.
- W3 - Some details are missing, hindering the readability of the paper.

---

> ### Author Rebuttal · Authors · 2025-07-31
>
> We sincerely thank the reviewer for the thoughtful and detailed feedback. We have carefully addressed all the comments by providing additional explanations and clarification.
>
> **W1: The connection between the forms in Eqs. 5 and 6 and the perturbation analysis is unclear.**
>
> **A**: The perturbation analysis in Sec. 3.2 is deeper analysis of the first item in Eqs.6 in Sec. 3.1.
>
> Sec. 3.1 (Eqs.6 and 6) highlights the need to match both soft label and image distributions. To further investigate soft label bias, Sec. 3.2 conducts a perturbation analysis showing that biases from both the distillation model and distilled images directly cause soft label mismatches. Importantly, these biases affect the soft labels directly and are not simply a consequence of image misalignment as described in Theorem 3.1 (see implication (iii), L174–L177). Then Sec. 3.3 introduces the ADSA module to jointly mitigate both types of soft label bias found in Sec. 3.2 in a unified manner.
>
> We will revise L166–L178 and the introduction to better highlight our focus on soft label alignment.
>
> **W2: The proposed method is disconnected from two different bias sources.**
>
> **A:** Our proposed method could mitigate the two bias sources (distilled images and distillation model) in soft label, which correspond to the soft label part in Th. 3.1.
>
> We have added two new experiments that explicitly evaluate our method's effectiveness against each individual bias.
>
> Exp1: Addressing bias from the distillation model (biased soft labels)
>
> |IF| biased img | biased label | method | IPC=1 | IPC=10 | IPC=50 |
> |-|-|-|-| - | - | - |
> | 10|N|Y|SRe2L| 16.6| 34.3| 50.6|
> ||N| Y|+ours| **19.0** | **41.4** | **62.6** |
> | 100|N| Y|SRe2L| 18.3| 24.4| 31.5|
> || N| Y|+ours| **21.8** | **36.9** | **57.6** |
>
> Exp2: Addressing bias in distilled images
>
> | IF| biased img | biased label | method | IPC=1 | IPC=10 | IPC=50 |
> | - | - | - | - | - | - | - |
> | 10| Y | N| SRe2L| 16.0 | 34.7 | 55.3|
> || Y  | Y|+ours| **23.7** | **40.2**  | **63.1** |
> | 100 | Y | N| SRe2L|15.6 | 22.9 | 54.8 |
> ||Y|Y|+ours| **21.4** | **33.2** | **58.4** |
>
> >  Note:
> >
> > 1. "Y" indicates the presence of bias; "N" indicates no bias. "IF" means Imbalance Factor.
> > 2. Biased img/label refers to one generated by a model trained on a long-tailed dataset.
>
> Results in Exp2 indicate that our method still outperforms the baseline even both the distilled images and soft labels are biased . This highlights the effectiveness of our method in mitigating the impact of image-induced bias.
>
> We do not adopt the baseline setting (biased images with unbiased labels) in Exp2 because our method relies on class-frequency information from a long-tailed distribution, which aligns with practical scenarios. However, in Exp2, this information is inconsistent with the model providing unbiased labels.
>
> A key contribution of our method is its ability to implicitly rectify both sources of bias on soft labels including distilled image bias and soft label bias in a unified manner.
>
> **W3: Some details are missing, hindering the readability of the paper.**
>
> A complete definition of all symbols and abbreviations will be provided, and the experimental setups from Sec. 3.2 and Sec. 4 will be further detailed in the appendix.
>
> **Q1: Can you include the intermediate steps when deriving the forms 1 and 2?**
>
> **A:** We will include a more detailed derivation for Forms (1) and (2) in Appendix A.1 of the revised version. Below we briefly outline the derivation of Form (1) for clarity.
>
> We first replace $p_{tr}$ with $p_{te}$ in the generalization bound from D3S [1]. Then, under the assumption of conditional feature consistency, $p_{tr}(x|y)=p_{te}(x|y)$, the test distribution can be rewritten as:
> $$
> p_{te}(x,y)=\frac{p_{te}(y)}{p_{tr}(y)}p_{tr}(x,y).
> $$
> Substituting this expression into the bound from D3S yields:
> $$
> l_{te} \leq l_{dd}+\frac{C}{2 \sqrt{2}} \sqrt{D_{K L}\left(\frac{p_{te}(y)}{p_{tr}(y)}p_{tr}(x, y) \| p_{dd}(x, y)\right)}
> $$
> Expanding the KL divergence term:
> $$
> \begin{aligned}
> & D_{K L}\left(\frac{p_{te}(y)}{p_{tr}(y)}p_{tr}(x, y) \| p_{dd}(x, y)\right)\\\\
> =& \int\frac{p_{te}(y)}{p_{tr}(y)}p_{tr}(x, y)\log \frac{p_{te}(y)p_{tr}(x,y)}{p_{tr}(y)p_{dd}(x,y)} dxdy\\\\
> =& \int\frac{p_{te}(y)}{p_{tr}(y)}p_{tr}(x, y)\log \frac{p_{tr}(x|y)}{p_{dd}(x|y)} dxdy + \int\frac{p_{te}(y)}{p_{tr}(y)}p_{tr}(x, y)\log \frac{p_{te}(y)p_{tr}(y)}{p_{tr}(y)p_{dd}(y)} dxdy\\\\
> =& \int p_{te}(y)p_{tr}(x|y)\log \frac{p_{tr}(x|y)}{p_{dd}(x|y)} dxdy+\int p_{te}(y)\log \frac{p_{te}(y)p_{tr}(y)}{p_{tr}(y)p_{dd}(y)} d\\\\
> =& \int p_{te}(y)D_{KL} \left(p_{tr}(x|y)\| p_{dd}(x|y) \right) dy+D_{KL}(p_{te}(y)\| p_{dd}(y))
> \end{aligned}
> $$
> Finally, we convert the integral form into an empirical summation form for practical implementation. This gives the decomposition into Form (1). Form (2) could be derived by applying the alternative decomposition $p(x,y)=p(y|x)p(x)$.
>
> **Q2: Identical samples could still minimize the second term in Eq 6.**
>
> **A:** Although identical samples can help reduce the second term, mismatched labels can lead to a significant increase in the first term.
>
> **Q3: The connection between the analysis in Sec. 3.2. and the insights drawn from Th 3.1. to be unclear.**
>
> **A:** We believe Q3 relates closely to W1, as both concern the connection between Theorem 3.1 and the perturbation analysis in Section 3.2. As noted in W1, Theorem 3.1 provides the theoretical foundation for both distilled images and soft label, while Section 3.2 offers empirical insights into soft label mismatch, while mainstream work focuses on distilled images.
>
> **Q3.1: Connection between configs in 3.2 and two forms in 3.1. Add config (4).**
>
> **A:** The three configss in Sec. 3.2 are designed to investigate the bias sources in the soft label distribution rather than distilled image distribution, and thus directly relate to the soft label part (first item) in form (2): $D_{KL}(p_{te}(y|x)\Vert p_{dd}(y|x))$.
>
> In fact, config 4 corresponds to the rightmost endpoint of all curves in configs 1-3, where the number of tail samples reaches 100. We will explicitly include config 4 in the revised version.
>
> **Q3.2：It would be interesting to see what happens to the overall performance (overall tail+head performance) given the derived generalization bound.**
>
> **A:** All experiments in Sec. 4 report the overall performance across all classes. Additionally, we provide a breakdown of performance across head, mid, and tail classes on ImageNet in Table 2, and a more fine-grained class-wise analysis on CIFAR-100 in Figure 3. These results show that our method improves both overall and tail-class performance.
>
> **Q3.3: L242-244. This finding contradicts the theory developed.**
>
> **A:** The finding in L242–244 provides a deeper investigation into the soft-label component of Th. 3.1. We identify two sources of soft-label mismatch - originating from both the distillation model and the distilled images.
>
> Most existing work in dataset distillation has primarily focused on improving image matching insteal of soft label. And different performance of baseline methods in Sec.4.1 also supports the image-matching component of Th. 3.1. However, this paper mainly focuses on the less-explored role of soft labels in dataset distillation.
>
> **Q3.4:  L212-214. We should draw that conclusion by comparing config 1 with config 3.**
>
> **A:** We will revise the statement to ensure greater clarity since config 1 shows better performance with biased images.
>
> **Q3.5: When solving Eq. 7. across the three configs, are you using the same $g$ network and only ablate the $f_{tr}$?**
>
> **A:** We do **not** use a shared or fixed $g$ across different configurations, and both $f_{tr}$ and its $g$ component are re-initialized and trained independently for each configuration. The $g$ network is introduced to describe the feature distribution matching in L192 more clearly.
>
> **Q4: Fig3. (b) The proposed method does not display a decreasing performance trend as the IF increases.**
>
> **A:** The issue stems from our default setting (Epoch=200) for SRe2L leading to suboptimal performance. We compared distilled data using models trained with different epochs on CIFAR10.
>
> Exp: Effect of Model Epoch on Performance with IF=10/100
>
> | Epoch | IF| Method | IPC=10 | IPC=50|
> | - | - | - | - |-|
> | 10 |10| SRe2L  | 39.6 | 55.0 |
> | | | +ours| 41.9 | 63.2 |
> | 30  | 10 | SRe2L  | 27.2  | 51.5 |
> |  | | +ours  | 26.2   | 50.0 |
> | 50 | 10 | SRe2L  | 24.6 | 46.8 |
> | |  | +ours  | 25.2 | 53.6|
>
> | Epoch | IF   | Method | IPC=10   | IPC=50   |
> | - | - | - | - | - |
> | 10    | 100  | SRe2L  | 21.6 | 23.6     |
> | | | +ours  | 33.0 | 37.9 |
> | 30    | 100  | SRe2L  | 28.1 | 38.2 |
> |  || +ours  | 32.0 | 59.1 |
> | 50 | 100  | SRe2L  | 23.9 | 40.3 |
> | | | +ours  | 29.1  | 53.8 |
>
> We observe that distilled performance first increases then drops with increasing distillation model epochs. The optimal epoch becomes larger as IF or IPC increases. This is because early-stage models yield richer soft labels[1], while [2–3] highlight the importance of proper epoch selection in trajectory matching.
>
> In the original setting, we use fixed epochs (200) for the model. As the IF increases, the optimal epoch shifts later, which explains the seemingly anomalous improvement in performance with higher imbalance.
>
> We revised our experimental setup by selecting the best-performing model from $\text{Epoch}\in\set{10,30,50}$.
>
> | tau\IF|10|50|100|500|1000|
> |-|-|-|-|-|-|
> |0| 32.6|28.2|29.2|23.9|21.9|
> |1|39.7|33.5|34.0|30.1|25.3|
> |1.5|40.6|33.3|34.4|28.8|28.2|
> |ours|**41.6**|**42.7**|**35.8**|**32.5**|**29.3**|
>
> We can observed that the previous anomalous performance trend disappears.
>
> **Typos：**
>
> **A:** We sincerely thank the reviewer for the careful feedback. We have corrected all identified issues in our internal revision.
>
> Refs:
>
> [1] Large Scale Dataset Distillation with Domain Shift, ICML, 2024

---

> ### Author Response · Authors · 2025-08-05
> **Kindly Request Feedback from Reviewer bfAk**
>
> Dear Reviewer bfAk,
>
> We sincerely appreciate the time and effort you devoted to reviewing our manuscript. In response to your thoughtful feedback, we have submitted a rebuttal with extensive experimental results addressing your concerns. The key updates include:
>
> - **Relationship between the Section 3.1 theory and Section 3.2 perturbation analysis:** Section 3.1 introduces a new theoretical framework for long‑tail data distillation, integrating both soft‑label matching and image matching. As prior work (e.g., SRe2L, GVBSM, EDC) focused primarily on image matching, our work is the first to investigate the role of soft‑label matching in this context. In Section 3.2, we show that biases in either the distilled images or the distillation model can lead to bias in soft labels.
> - **Effectiveness of ADSA for handling both bias sources:** We designed experiments demonstrating that ADSA effectively mitigates bias originating from the distillation model and from the distilled images themselves.
> - **Additional proof for the theoretical derivation:** We now include a detailed proof of Form 1, and clarify that Form 2 can be derived by a parallel argument. If desired, we are willing to provide the full proof of Form 2 during the discussion period.
> - **Performance under extreme imbalance factors:** We investigated the behavior under extreme imbalance and identified that fixing the distillation‑model epoch leads to improved performance. By selecting the optimal epoch, we resolved the performance drop and achieved an overall gain.
> - **Fixes to typos in the revised version:** We have improved clarity throughout the revised manuscript, resolving previous ambiguities and correcting typos.
>
> We hope that these clarifications and additional experiments help address your concerns. We welcome any further feedback or questions. Thank you again for your invaluable feedback and for considering our response during the rebuttal process.
>
> Sincerely,
>
> The Authors

---

> ### Comment · Reviewer_bfAk · 2025-08-05
>
> I thank the authors for their rebuttal and the additional experiments provided. After reading the rebuttal, I still have some follow-up questions and comments:
>
> ** W1 **
>
> > The perturbation analysis in Sec. 3.2 is deeper analysis of the first item in Eqs.6 in Sec. 3.1.
>
> This has to be explicitly stated at the beginning of 3.2. Beyond that, I would ideally have expected a brief explanation on why the perturbation analysis is conducted based on Eq 6. and not Eq. 5., there has to be a reason behind this decision which would be interesting to document. The latter would further strengthen the connection between the theory and the perturbation analysis.
>
> > Sec. 3.1 (Eqs.6 and 6) highlights the need to match both soft label and image distributions.
>
> “(Eqs.6 and 6) “ is this a typo? If not, can you please explain the equation(s) you are referring to?
>
> When looking at the first term of Eq 6. I understand the need to match the soft-label distribution, which manifests as the posterior distribution in the first term of Eq. 6.
>
> When looking at the first term of Eq 6. I do not understand the need to match the image distributions. Can you please elaborate how you deduce the “need to match” both soft-label and image distributions from the first term in Eq. 6 or should we look at the second term of Eq. 6 as well?
>
> > To further investigate soft label bias, Sec. 3.2 conducts a perturbation analysis showing that biases from both the distillation model and distilled images directly cause soft label mismatches.
>
> You have established that in theory (first term Eq. 6), that soft-label mismatch can negatively affect the final performance. Given the title of S.3.2 and L180-181, you are arguing that both the soft-labelling model (i.e., DM2) and the model used to distill the images (i.e., DM1) introduce bias in the soft-labels (i.e., first term Eq. 6).
>
> Do I understand this correctly?
> If yes, then you need to clarify when the soft-labels refer to the posterior of the first term Eq. 6 and when to the soft-labelling model DM2.
> Given that the performance depends on both terms of Eq. 6, the perturbation analysis investigates the effect on the two pathways jointly on the soft-labels (first term of Eq. 6) and the images (second term of Eq. 6). Based on that I think it is important to clarify that the dual nature of influence affects both these terms.
> Next, it is important to clarify that the proposed method (ADSA) addresses the dual biases only in the soft-labels (first term of Eq. 6). In other words, I would appreciate some more transparency on what the analysis 3.2. Investigates and how ADSA relates to it.
>
>
> > Importantly, these biases affect the soft labels directly
>
> Can you please elaborate why this is the case? (also related to my previous point on attributing **both** pathways to the first term of Eq. 6).
>
> > Importantly, these biases affect the soft labels directly and are not simply a consequence of image misalignment as described in Theorem 3.1 (see implication (iii), L174–L177).
>
> I understand that these biases affect the performance and therefore both terms of Eq. 6. I do not understand the reason for referencing point (iii) in this context.
>
> > We will revise L166–L178 and the introduction to better highlight our focus on soft label alignment.
>
> I consider the L166–L178 a key part of your work. Could you please provide the full revision?
>
> ** W2 **
>
> > Our proposed method could mitigate the two bias sources (distilled images and distillation model)
>
> You convincingly demonstrate in the Exp1 and Exp2 that bias in both DM1 and DM2 affect the final performance which ADSA improves upon.
>
>  However, addressing W2 would require providing the causal roadmap going from analyzing the Th 3.1. (L166:L178), to the perturbation analysis in S.2 and finally the proposed method ADSA. To this end I still find the connection loose. Alternatively, you will need to clearly specify which part of the analysis informed the development of the method.
>
>
> ** Q1 **
>
> > We will include a more detailed derivation for Forms (1) and (2) in Appendix A.1 of the revised version.
>
> Thank you.
>
> ** Q2 **
>
> > Although identical samples can help reduce the second term, mismatched labels can lead to a significant increase in the first term.
>
> I understand this, but your statement in L174-L176  “The second term
> in both Eq.(5) and Eq.(6) suggests that, for each class, the feature distribution in the synthetic dataset should match that of the original data to achieve optimal performance” refers to the second term of Eq. 6. in isolation of the first term of Eq. 6. which is not correct and introduces confusion. This reinforces my insistence on looking at the full revision of L166-L178 prior to my decision.

---

> > ### Comment · Reviewer_bfAk · 2025-08-05
> >
> > ** Q3 **
> >
> > > As noted in W1, Theorem 3.1 provides the theoretical foundation for both distilled images and soft label
> >
> > Here you refer to the first and second terms of Eq. 6?
> >
> > > As noted in W1, Theorem 3.1 provides the theoretical foundation for both distilled images and soft label, while Section 3.2 offers empirical insights into soft label mismatch, while mainstream work focuses on distilled images.
> >
> > Related to my previous point under W1, isn’t Section 3.2 offering insights jointly on both soft label and image mismatches? If not, can you elaborate?
> >
> > ** Q3.1 **
> >
> > > The three configss in Sec. 3.2 are designed to investigate the bias sources in the soft label distribution rather than distilled image distribution, and thus directly relate to the soft label part (first item) in form (2):
> >
> > Do you mean that you investigate the bias sources in the soft label independently from those in the distilled image distribution? To my current understanding the analysis in S.3.2 and therefore the configs analyze the effect of the dual biases in both soft labels and distilled image distribution (i.e. both terms of Eq. 6.).
> >
> > > In fact, config 4 corresponds to the rightmost endpoint of all curves in configs 1-3, where the number of tail samples reaches 100. We will explicitly include config 4 in the revised version.
> >
> > Thank you for the clarification. Including the config 4 will help readers better ground the effect of all configs.
> >
> > ** Q3.2 **
> >
> > > All experiments in Sec. 4 report the overall performance across all classes
> >
> > Ok, but the Q3:X questions target S.3.2 specifically and therefore targets the configs1-3.
> >
> > ** Q3.3 **
> >
> > I follow up on this, if needed, after you have answered the previous questions.
> >
> > ** Q3.4 **
> >
> > > We will revise the statement to ensure greater clarity
> >
> > Could you please provide the revised statement?
> >
> > ** Q.3.5 **
> >
> > Consider including the explanation either in the main text (potentially as a footnote) or in the appendix.
> >
> > ** Q4 **
> >
> > Thanks for the clarification.

---

> > > ### Author Response · Authors · 2025-08-06
> > >
> > > We sincerely thank the reviewer for the thorough reading and the careful, insightful questions. We fully understand your concerns and, after carefully reviewing your comments, we identified several key points in the original manuscript that may have led to misunderstanding:
> > >
> > > - The experiment in Section 3.2 is intended to demonstrate two aspects:
> > >   (1) both mismatched soft labels and mismatched distilled images can lead to performance degradation (Th. 3.1);
> > >   (2) biased distilled images and a biased model used for labeling can result in biased soft labels, which in turn degrade performance (first item in Eq.6).
> > > - We acknowledge that the manuscript lacked emphasis on the fact that our core focus is on biased soft labels.
> > > - The phrase *"bias from distillation model"* may cause confusion, as the distillation model can also be used to generate the distilled images. To improve clarity and precision, we will revise it to *"bias from labeling model."*
> > > - "labeling model"/"DM2" could both refer to the soft label part in Eq. 6 and specific "bias from labeling model" in Sec. 3.2. This is because "labeling model" only affects the performance by introducing bias in soft label, not like distilled images could affect the performance by itself or introducing bias in soft label.
> > >
> > > In the following response: (1) we first present the revised version of the original text to better highlight our main focus; (2) we then address your questions point by point, in particular, **clarifying the connections between Sections 3.1, 3.2, and 3.3, and providing a clear roadmap of our analysis**.

---

> ### Author Response · Authors · 2025-08-06
>
> **revised text**
>
> > L166-L178
>
> The proof is provided in Appendix A.1. Under the relaxed assumption, the two resulting bounds involve additional $p_{te}$ terms compared to Eq. (4), leading to two key insights: (i) The first terms in both Eq. (5) and Eq. (6) demonstrate that the label distribution in the distilled dataset should match the label distribution in the test set; (ii) The second terms in both equations show that the distribution of distilled images should align with that of the training images. Most existing works focus on improving the performance of the distilled dataset by addressing insight (ii). For instance, EDC[1] incorporates both global and class-aware feature matching between distilled and original images. However, the widely adopted and empirically effective soft-labeling technique remains underexplored. Moreover, soft labels are highly generalizable and can be seamlessly integrated into various distillation pipelines. Therefore, we focus more on the label-related terms in insight (i).
> The first term in Eq. (5) suggests a natural experimental setup: each class in the distilled dataset should contain the same number of samples as in the test set, resulting in a balanced class distribution. Furthermore, the first term in Eq. (6) indicates that the posterior distribution of soft labels, $p_{dd}(y|x)$, should align with the label distribution in the test set. In Section 3.2, we show that the commonly adopted soft-labeling strategy introduces bias in long-tailed dataset distillation, which leads to performance degradation. We further identify two primary sources of this bias inherent in existing soft-labeling techniques.
>
> > L180
>
> In this section, we first empirically validate the theoretical insights (i) and (ii) presented above. We then analyze the soft labels to identify two key factors that undermine insight (i) in the context of long-tailed dataset distillation.
>
> > L209-L235
>
> As shown in Figure 2, panel (a), config 1 (using only imbalanced images), config 2 (using only an imbalanced distillation model for labeling), and config 3 (both imbalanced) all experienced a performance drop compared to config 4 (both balanced). This validates the theory presented in Section 3.1, which states that both biased distilled images and soft labels lead to performance degradation. Furthermore, we observed that config 2, which used an imbalanced model for labeling, showed a greater performance decrease than config 1. Panel (b) illustrates the entropy of the soft labels, showing that a more imbalanced dataset leads to higher entropy, which indicates a lack of class-discriminative information in the soft labels[2].
>
> To further analyze the respective contributions of imbalanced distilled images and imbalanced distillation models to the final soft labels, we visualize the per-class confidence. We use the term *labeling model* to refer to the distillation model used to generate soft labels for the distilled images. Panels (c) and (d) show that both imbalanced synthetic images and imbalanced labeling models lead to soft labels that are overconfident for head classes and underconfident for tail classes. Notably, even in config 1 where a balanced distillation model is used, its predictions are biased due to the imbalanced distilled images, and we refer to this as *bias from distilled images*. In contrast, the bias observed in config 2 stems from the prediction bias within the labeling model itself, which we term *bias from the labeling model*. As a result, the evaluation model trained on such biased soft labels from these two sources learns incorrect class probabilities, thereby inheriting this bias.
>
> [1] Elucidating the Design Space of Dataset Condensation, NeurIPS, 2024
>
> [2] A Label Is Worth a Thousand Images in Dataset Distillation, NeurIPS, 2024

---

> ### Author Response · Authors · 2025-08-06
>
> **W1:**
>
> > Focusing on first item in Eq. 6 should be explicitly stated at the beginning of 3.2.
>
> We have added the following statement at the beginning of L180:
>
> "In this section, we first empirically validate the theoretical insights about soft labels and distilled images presented in Th. 3.1. We then focus on soft labels and identify two key factors that lead to biased soft labels, which in turn degrade the performance of the distilled dataset."
>
> > Reason for chosing Eq. 6 over  Eq. 5
>
> In fact, we focus on the label-related terms rather than the distilled images. The label term in Eq. (5) simply reflects the standard experimental setting in long-tailed dataset distillation, where each class is allocated the same number of distilled images. Therefore, our analysis centers on the label term in Eq. (6), aiming to investigate the potential causes of soft label mismatch.
>
> There are two main reasons for focusing on the label-related terms rather than the distilled images. First, many existed works have already focused on improving image feature alignment (e.g., SRe2L, GVBSM, EDC, DREAM), while soft-labeling remains underexplored. In fact, soft labels can bring significant performance improvements to distilled datasets.
> Second, soft labels are a highly generalizable technique that can be applied across a wide range of distillation methods. Our ultimate goal is to develop a training-free solution that avoids additional architectural design or feature matching mechanisms, making the final approach both efficient and broadly applicable. Therefore, we choose to focus our study on soft labels.
>
> > When looking at the first term of Eq. 6. I do not understand the need to match the image distributions.
>
> Sorry, it should refer to Eqs. (5) and (6). The first term in Eq. (6) corresponds to matching soft labels, while the second term relates to matching distilled images. We consider both terms in our analysis, and subsequently focus on the soft label matching in depth.
>
> > Clarification on when the soft-labels refer to the posterior of the first term Eq. 6 and when to the soft-labelling model DM2.
>
> You are absolutely correct in your understanding, and we appreciate your insightful summary. We recognize that this point is central to potential confusion, and we will clarify it accordingly in the revision. Below, we provide detailed clarifications:
>
> * In dataset distillation, the same model is typically used both to generate distilled images and to produce soft labels. However, in our analysis experiment, to better isolate and understand the different sources of bias, we decouple these two roles. Specifically, we refer to the model responsible for generating soft labels as DM2, and the model used to produce the distilled images as DM1.
>   * In L224–L240, we use the term "bias from distillation model" to specifically refer to the model used for labeling, which may cause ambiguity. We will revise this terminology to "labeling model" for clarity. We will unifying the statement of "DM2" and "labeling model" in revised paper.
> * As shown in Figure 2(a), both the labeling model and the distilled images contribute to performance degradation. Since the labeling model directly determines the soft labels, and the images affect both feature alignment and soft label generation, this empirically supports both terms in Eq. (6).
>   * "labeling model"/"DM2" could both refer to the soft label part in Eq. 6 and specific "bias from labeling model" in Sec. 3.2. This is because "labeling model" only affects the performance by introducing bias in soft label, not like distilled images could affect the performance by itself or introducing bias in soft label.
> * In Figures 2(c) and 2(d), we further analyze soft label bias. The soft label $\hat{y} = f(x)$ depends on both the labeling model $f$ and the distilled image $x$. While it is natural that a biased labeling model $f$ leads to biased $\hat{y}$, we additionally observe that biased $x$ (from the distilled image set) can also induce bias in $\hat{y}$, even when $f$ is balanced. This reveals an indirect but significant influence path: distilled images → soft labels → performance, highlighting that the impact of images is not only direct (second term of Eq. 6), but also indirect (via the first term).
> * The proposed ADSA module is designed to correct the bias in the soft labels $\hat{y}$ caused jointly by both $f$ and $x$. Specifically, ADSA is used to calibrate the imbalanced confidence induced by biased distilled images and biased labeling model. In other words, ADSA targets the first term of Eq. (6), but does not address the direct influence of $x$ on performance (i.e., the second term). This distinction will be clearly stated in the revised version.

---

> ### Author Response · Authors · 2025-08-06
>
> > Clarifying the attribution of dual bias sources to the soft label term in Eq. (6)
>
> Figures 2(c) and 2(d) demonstrate that both the distilled images and the labeling model influence the confidence of the soft labels. These biased soft labels, in turn, lead the evaluation model trained on them to inherit the bias, ultimately resulting in degraded performance.
>
> > Reason for mentioning implication (iii)
>
> We referenced implication (iii) only to provide additional context for image misalignment. The mention of (iii) is not essential to our argument and can be safely omitted for clarity.
>
> > Provide the full revision about L166–L178.
>
> We have provided above.
>
> **W2**
>
> > Providing the causal roadmap going from analyzing the Th 3.1. (L166:L178), to the perturbation analysis in Sec. 2 and finally the proposed method ADSA.
>
> This is a crucial point, and we appreciate the request. **Below, we provide a clear reasoning path that connects our theoretical formulation to the perturbation analysis and the final proposed solution**:
>
> * Sec. 3.1: Th. 3.1 suggests that both soft label mismatch and distilled image distribution mismatch contribute to performance degradation.
> * Sec. 3.2:
>   * Both the labeling model(representing soft label here) and distilled images affect final performance (Fig. 2(a)), validating the two terms in soft label part in distilled images part in Th. 3.1.
>   * Both the labeling model and the distilled images lead to biased soft labels (Fig. 2(c) and 2(d)), which degrade performance. This supports a more fine-grained interpretation of the first term in Eq. 6.
> * Sec 3.3:
>   * We propose ADSA, a module designed to mitigate the bias in soft labels caused by both the labeling model and the distilled images.
> * **Summary of the bias sources and performance drop**:
>   * Labeling model → biased soft label matching → performance degradation
>   * Distilled images → biased soft label matching + biased image matching → performance degradation
>   * we focus on solving biased soft label matching by ADSA
>
> **Q2**
>
> > Statement in L174-L176 refers to the second term of Eq. 6. in isolation of the first term of Eq. 6.
>
> While optimizing the second term of Eq. (6) (i.e., matching the feature distribution of distilled images) is important, achieving optimal distillation performance requires both terms in Eq. (6) to be well satisfied.
>
> **Q3**
>
> > Here you refer to the first and second terms of Eq. 6?
>
> Yes.
>
> > Isn’t Section 3.2 offering insights jointly on both soft label and image mismatches
>
> In Fig.2 (a), yes. And then in Fig. 2(c) and (d), we focused on how distilled images and labeling model contribute to soft label mismatches.
>
> **Q3.1**
>
> > Clarification on whether section 3.2 separately or jointly analyzes biases in soft labels and distilled images
>
> We acknowledge that the original text may have caused confusion on this point. To clarify:
>
> - Figure 2(a) illustrates the joint impact of both the labeling model and the distilled images on the final performance.
> - Figures 2(c) and 2(d) further show that both factors individually contribute to biased soft labels, which in turn degrade performance.
>   Thus, Section 3.2 analyzes both the separate contributions of each component to soft label bias, and their joint effect on the overall performance.
>
> **Q3.2**
>
> > Report the overall performance in Sec. 3.2
>
> Understood. We report the overall accuracy by varying the number of images in tail classes.
>
> |  #image in tail classes |    5 |   10 |   15 |   20 |   40 |   60 |   80 | 100  |
> | -------: | ---: | ---: | ---: | ---: | ---: | ---: | ---: | ---- |
> | config 1 | 0.50 | 0.53 | 0.53 | 0.61 | 0.65 | 0.62 | 0.64 | 0.69 |
> | config 2 | 0.48 | 0.48 | 0.52 | 0.50 | 0.57 | 0.59 | 0.62 | 0.69 |
> | config 3 | 0.46 | 0.48 | 0.49 | 0.55 | 0.65 | 0.64 | 0.67 | 0.69 |
> | config 4 | 0.69 | 0.69 | 0.69 | 0.69 | 0.69 | 0.69 | 0.69 | 0.69 |
>
> The results show that both a biased labeling model and biased distilled images lead to overall performance degradation.
>
> **Q3.4**
>
> > Could you please provide the revised statement?
>
> We have provided the revised statement above. (L209-L235)
>
> **Q3.5**
>
> > Consider including the explanation about $g$ network either in the main text (potentially as a footnote) or in the appendix.
>
> We will add explanation in the main text.
>
>
> We sincerely appreciate your pursuit of rigorous logic, which has been truly beneficial to us. Your comments have prompted us to identify potential sources of confusion in the paper and have helped us further strengthen its logical flow. We genuinely hope that our response addresses your concerns, and we warmly welcome any further discussion or feedback.

---

> > ### Comment · Reviewer_bfAk · 2025-08-06
> >
> > I thank the author for their continual effort to provide clarifications. I appreciate acknowledging the initial sources of confusion and working towards clarifying them. I have few follow-up comments that I hope you can address.
> >
> > I would also like to kindly ask the author(s) to remain as concise and to the point as possible to facilitate our communication.
> >
> > ** Revised L166-L178 **
> >
> > The revised analysis now groups the terms of Eq 5. and Eq 6. into label-based and image-based. Although the analysis is now more concise I think jointly analysing conditional and unconditional terms can be problematic/confusing. For example:
> >
> > > (i) The first terms in both Eq. (5) and Eq. (6) demonstrate that the label distribution in the distilled dataset should match the label distribution in the test set;
> >
> > The explanation is accurate for the first term of Eq. 5. but not for Eq. 6. In fact, the first term of Eq. 6. requires matching label distribution conditioned of the images. Please revise the statement or split it into two to ensure correctness.
> >
> > The same argument applies for the point (ii) of the revised L166-178.
> >
> > * * Newly introduced statement in L180 **
> >
> > The provided statement does not address my initial concern. The reader (including me) still has to guess which equation(s) and term(s) to look at.
> >
> > My initial concern was that S.3.2. fails to connect the analysis explicitly to an equation (presumably Eq. 6) and term (presumably first term) to ground the understanding of the configurations and the overall perturbation analysis.
> >
> > ** Revised L209-L235 **
> >
> >  > This validates the theory presented in Section 3.1, which states that both biased distilled images and soft labels lead to performance degradation
> >
> > Based on your previous response, I am assuming you refer to the two terms of Eq. 6. I fail to see how the results of the perturbation analysis in S.3.2. validate the Th.3.1. developed in S.3.1.
> >
> > My understanding is:
> >
> > - DM1 directly controls the mismatch in the images and indirectly controls the mismatch in the soft-labels doing the soft-labeling on mismatched images, impacting **both** point (i)  (i.e., first term of Eq. 6.) and point (ii) (i.e., second of Eq. 6.).
> >
> > - DM2 directly controls the mismatch in the soft-labels which impacts **only** point (ii) (i.e., second of Eq. 6.)
> >
> > Based on that:
> >
> > - config1 negatively impacts **both** point (i) and point (ii)
> >
> > - config2 negatively impacts **only** point (i)
> >
> > - config3 negatively impacts **both** (i) and (ii)
> >
> > - config4 is an oracle with optimal handling of both points (i) and (ii)
> >
> > Empirically validating Th. 3..1 would require having another config5 (potentially unattainable in practice) such that:
> >
> > -config5 negatively impacts **only** point (ii)
> >
> > and show that config4 (i.e., the oracle) is better than both config2 and config5.
> >
> > At its current state you have verified that negatively impacting point (i) and negatively impacting the combination of point (i) and point (ii) degrade the performance. Although an interesting finding, the evidence is not sufficient to verify the Th3.1.
> >
> > On the other hand, the fact that both config1 and config3 outperform config2 (see Fig 2 panel a) calls for a deeper discussion in regards to Th3.1. as negatively affecting point (i) performs worse than negatively affecting both point (i) and point (ii).
> >
> > A potential explanation to this behavior could be that config2 affecting **only** point (i) has a more severe impact on the bound (Eq. 6). compared the effect of config1 and config3, despite the later two configs negatively impacting both point (i) and point (ii).
> >
> > On that basis, I stand by my initial view that the perturbation analysis is disconnected from the theory and the proposed method (ADSA) disconnected from the perturbation analysis and from the fact that there is a dual source of bias in soft-labels.
> >
> > ** On the causal roadmap between theory, perturbation analysis and proposed method **
> >
> > > In other words, ADSA targets the first term of Eq. (6), but does not address the direct influence of $x$ on performance (i.e., the second term).
> >
> > Thank you for the clarification.
> >
> > I now see the connection between the proposed method (ADSA) and Th 3.1., consider providing such explicit connection between ADSA and Eq. (6) in S.3.3.
> >
> > Note that such connection does only partly verify Th 3.1. (i.e., from the standpoint of soft-labels) and this has to be explicitly stated in the text to avoid confusion.
> >
> > I still fail to see how the perturbation analysis has informed any of your methodological decisions or theoretical insights. Please refer to my points raised under ** Revised L209-L235 **.

---

> > > ### Comment · Reviewer_bfAk · 2025-08-06
> > >
> > > ** Q2 **
> > >
> > > > While optimizing the second term of Eq. (6) (i.e., matching the feature distribution of distilled images) is important, achieving optimal distillation performance requires both terms in Eq. (6) to be well satisfied.
> > >
> > > Ok, but my initial point was on the incorrect claims of point (iii) which isolated the second term Eq. 6., your (two) answers so far made arguments using both terms of Eq. (6).
> > >
> > > Given that the point (iii) has been removed and the L166-L178 significantly revised, I will assume you understood my initial point on the incorrectness of point (iii).
> > >
> > >
> > > ** The overall performance **
> > >
> > > Thank you for providing the overall performance for the experiment in S.3.2. If I understand correctly, these closely follow the trend found in the original Fig 2. panel a.
> > >
> > > Concluding, I still think that there is either a missing (i) crucial connecting link between Th3.1./ADSA and the perturbation analysis/dual nature of bias or (ii) placing of the perturbation analysis/dual nature of soft-label bias and its insights in the right context.

---

> ### Author Response · Authors · 2025-08-07
>
> We sincerely appreciate your insightful analysis, which motivated us to conduct additional experiments and further clarify the connections between different sections of the paper. In particular, your comments prompted us to enhance the analysis and completeness of the experiments in Section 3.2.
>
> **Jointly analysing conditional and unconditional terms can be problematic/confusing**
>
> We will revise the statement to ensure concise:
>
> The proof is provided in Appendix A.1. Under the relaxed assumption, the two resulting bounds introduce additional terms compared to Eq. (4), leading to several key insights regarding the desired properties of the distilled dataset:
> (i) The first term in Eq. (6) suggests that the learned posterior distribution $p_{dd}(y|x)$ from a model trained on $D_{dd}$ should align with the label distribution in the test set.
> (ii) The second term in Eq. (6) highlights the importance of aligning the feature distribution of the synthetic dataset with that of the original training data to ensure optimal performance. Additionally, the second term in Eq. (5) refines this insight by emphasizing a more fine-grained, per-class alignment due to the long-tailed assumption $p_{tr}(x|y) = p_{te}(x|y)$.
> (iii) The first term in Eq. (5) further suggests a natural experimental setup by adopting a balanced class distribution in the distilled dataset, where each class contains the same number of samples as in the test set.
> While most existing works aim to improve distillation performance by focusing on insight (ii), such as EDC which enforces both global and class-aware feature matching corresponding to the second terms in Eq. (6) and Eq. (5), the widely used and empirically effective soft-labeling technique remains relatively underexplored. Notably, soft labels are highly generalizable and can be seamlessly integrated into various distillation pipelines.
> Therefore, this work primarily focuses on insight (i). In Section 3.2, we demonstrate that conventional soft-labeling strategies introduce bias in long-tailed dataset distillation, resulting in a mismatch in the first term of Eq. (6) and subsequent performance degradation. We further identify two main sources of this bias that are inherent in existing soft-labeling methods.
>
>
>
> **New statement in L180 is insufficient, readers still has to guess which equation(s) and term(s) to look at**
>
> L180:
>
> In this section, we first empirically validate that mismatches in soft labels (first term in Eq. 6) and distilled images (second term in Eq. 6) under long-tailed distributions lead to performance degradation. We then identify two key factors that distort the distribution of soft labels.
>
>
>
> **Empirically validating Th. 3.1 would require having another config5 to prove only image mismatch will influence performance**
>
> Understood. We conducted a new experiment to verify the influence of only distilled image mismatch. We replaced the soft labels with one-hot labels. Athough this results in a slight performance drop compared to using soft labels, but we can remove the influence of soft label bias induced from distilled images.
>
> Experiment was conducted on CIFAR100 using SRe2L (same with config 1/2/3).
>
> |  Number of Tail Images |     5 |    10 |    15 |    20 |    40 |    60 |    80 | 100   |
> | --------: | ----: | ----: | ----: | ----: | ----: | ----: | ----: | ----- |
> | Test Acc. | 20.96 | 22.53 | 24.31 | 22.93 | 25.88 | 25.67 | 26.41 | 25.78 |
>
> Similar to config 1 (which uses imbalanced images), we observe a rapid performance increase before epoch 20, followed by noticeable fluctuations. This result supports the implication of the second term in Eq. (6): long-tailed data leads to biased distilled images, ultimately causing performance degradation.
>
> We agree with your observation. The fact that both config1 and config3 outperform config2 calls for a more in-depth and fine-grained analysis of the first term in Eq. (6). For instance, while both a biased model $f$ and biased input $x$ can contribute to biased predictions, their effects may not follow a simple monotonic relationship, for example, increasing the bias in $f$ or $x$ does not necessarily result in increasingly biased soft label.

---

> ### Author Response · Authors · 2025-08-07
>
> **How the perturbation analysis has informed your methodological decisions**
>
> * Insight 1:
>   * Perturbation analysis: Biased labeling model leads to worse performance than biased distilled images.
>   * Method: Soft label calibration may be more important than improving image matching, especially considering limited exploration of soft labels in long-tailed dataset distillation.
> * Insight 2:
>   * Perturbation analysis: Biased distilled images also lead to biased soft label.
>   * Method: We should take distilled images into acount during the process of soft label calibration. Thus we compute the class-wise average soft label across all distilled images as $p(\bar{y}=i|x;\tau)=E_{x\sim D_i}[p(y=i|x;\tau)]$, where $D_i$ denotes the set of distilled images labeled as class $i$. (L268-L270, Section 3.3)
> * Insight 3:
>   * Perturbation analysis: Both the biased labeling model and biased distilled images contribute to final soft label jointly.
>   * Method: Instead of calibrating the labeling model and distilled images separately, we aim to calibrate the final soft label directly, due to the complex interplay between the two sources. Proper calibration of the soft label can jointly mitigate both sources of bias.
>   * Experiment: The Exp1 and Exp2 in rebuttal process has shown that the ADSA could mitigate the biases from both two sources.
>
> **Conclusion**
>
> Connection between each section:
>
> * Section 3.1 and Section 3.2: Perturbation analysis(with new one-hot experiment) validates the first item and second item in Eq. 6. And in Section 3.2 we identify two bias sources in soft label (first item in Eq. 6) further.
> * Section 3.2 and Section 3.3: ADSA leverages distilled images to help mitigate the bias introduced by them. And ADSA in Section 3.3 could mitigate both sources of soft label bias identified in Sectino 3.2.
> * Section 3.1 and Section 3.3: ADSA aims to better align the first term in Eq. (6), which is orthogonal to the second term. The latter can be addressed using various image matching strategies commonly adopted in dataset distillation, e.g. SRe2L, GVBSM, EDC.
>
> Once again, we sincerely thank the reviewer for the time and effort devoted to the detailed and thoughtful questions. We truly hope that our responses have addressed your concerns and provided the necessary clarity.

---

> ### Comment · Reviewer_bfAk · 2025-08-08
>
> I thank to the authors for the engagement. Despite my initial concerns, the authors have made a substantial effort in the rebuttal and the discussion that followed, providing complementary theoretical derivations and additional experiments. On balance, the strengths of the paper outweigh minor reservations, and I will therefore be raising my score.

---

> > ### Author Response · Authors · 2025-08-08
> >
> > We sincerely thank the reviewer for your support of our work. We truly appreciate your thoughtful and rigorous suggestions during the rebuttal and discussion phases, which have deeply inspired us. We will strive to carry forward this spirit of rigor in our future research.

---

### Author Response · Authors · 2025-08-05
**Response to all reviewers**

We sincerely thank all reviewers for their thorough and constructive feedback. We are encouraged by the recognition of our theoretical analysis on the upper bound for long-tailed dataset distillation, the emperical identification of two key sources of soft-label bias, and the effectiveness of the proposed ADSA module.

# Key Improvements and Clarifications

1. Technical Novelty and Component Analysis

   - Provided detailed explanations connecting the theory in Section 3.1 with the three configurations in Section 3.2, as well as the final method. The last two focus on the two key bias sources in soft label distribution in Section 3.1.

   - Demonstrated that the proposed ADSA module effectively mitigates bias introduced by both the distilled images and the distillation model through ablation studies.

   - Theoretically proved that ADSA reduces to the dataset distillation baseline on balanced datasets.

2. Extended Evaluation and Analysis

   - Provided a detailed comparison between our method and the newly introduced resampling-based baseline.

   - Clarified that the accuracy improvement under extreme imbalance factors (IFs) in Figure 3(b) is attributed to the fixed distillation model epoch.

   - Reproduced the missing experimental results for LAD baseline and reported the updated results to ensure completeness.

   - Conducted multiple runs with variance to demonstrate the stability of our method.

   - Provided GPU memory and runtime overhead under various parameter settings for both the baseline methods and our lightweight module.

3. Other Clarifications
   - Provided a detailed proof of the loss upper bound.

Please see our detailed responses under each individual review. We sincerely appreciate your feedback and welcome any further questions.

---

### Comment · Area_Chair_Xqpi · 2025-08-05

Dear Reviewers,

Please read the other reviews and the author's response, and start a discussion with the authors promptly to allow time for an exchange.

Your AC

---

### Note · Authors · 2025-08-13

We sincerely thank all reviewers, ACs, and SACs for their valuable time in carefully evaluating our work. This paper presents an in-depth study of the critical yet underexplored problem of long-tailed dataset distillation. We theoretically and experimentally analyze why existing methods degrade on long-tailed data, identify soft-label mismatch as a key cause, and propose **ADSA**, a lightweight module that consistently improves performance across diverse experimental settings and baselines. We are greatly encouraged that all reviewers recognized our responses throughout the rebuttal and discussion periods.

During the rebuttal and discussion phases, we carefully considered all feedback. Our main responses to each reviewer’s key concerns are as follows:

- **Reviewer bfAk:** We are grateful to the reviewer for requesting clarification on the connections between different sections. Detailed explanations and additional experiments were provided to strengthen the link between the theory, analysis, and method. The observation that performance improves as the IF increases was clarified.
- **Reviewer GTsB:** We greatly appreciate the reviewer’s constructive comments on adopting a new baseline and verifying the handling of biases. In response, we ran the resampling baseline, which demonstrated consistent performance gains of our method; conducted additional experiments to confirm ADSA’s effectiveness in mitigating both biases; and clarified points of confusion in the experimental setup.
- **Reviewer Prxx:** We acknowledge the reviewer’s request for additional details, including the missing LAD results, hardware usage information, and multiple-run results of our method. Comprehensive experiments were conducted and the results were reported in detail.
- **Reviewer cicZ:** We appreciate the reviewer’s thoughtful questions on the effectiveness of addressing dual bias, computational resources, the Fig. 3(b) anomaly, and performance on balanced datasets. Additional experiments were performed and thorough explanations were provided for each point.

In addition, detailed responses were provided to all other questions from the reviewers, including explaining terms and adding intermediate steps in the proofs. All suggestions from the reviewers will be thoroughly incorporated into the revised version of the paper. We thank all reviewers for their collective effort and constructive feedback, which have significantly contributed to improving this work.

---

### Decision · Program_Chairs · 2025-09-17

**Decision:**

Accept (poster)

**Comment:**

This paper presents a new method for long-tailed dataset distillation, which has not been well attempted in previous works. Specifically, the paper generates soft labels for compressed data points using post-hoc logit adjustment. Experiments demonstrate the effectiveness of the method.

After carefully reviewing the paper, the AC find that the paper does not propose new dataset compression approaches, and the generation of soft labels using logit adjustment is not new in long-tail learning. These can be potential weaknesses of the paper.

After rebuttal, the paper received four positive ratings. All reviewers acknowledged the paper's contributions and expressed satisfaction with its technical merits.

The AC slightly tends to concur with the reviewers' assessments and recommends acceptance. not new.